# Regulation of sleep homeostasis by sexual arousal

Esteban J Beckwith, Quentin Geissmann, Alice S French, Giorgio F Gilestro*

Department of Life Sciences, Imperial College London, London, United Kingdom

**Abstract** In all animals, sleep pressure is under continuous tight regulation. It is universally accepted that this regulation arises from a two-process model, integrating both a circadian and a homeostatic controller. Here we explore the role of environmental social signals as a third, parallel controller of sleep homeostasis and sleep pressure. We show that, in *Drosophila melanogaster* males, sleep pressure after sleep deprivation can be counteracted by raising their sexual arousal, either by engaging the flies with prolonged courtship activity or merely by exposing them to female pheromones.
DOI: https://doi.org/10.7554/eLife.27445.001

## Introduction

The two-process model for regulation of sleep, first postulated by Borbély in 1982, is still considered the most accurate conceptual framework to describe how sleep pressure builds and dissipates along the day (*Borbély, 1982*). According to the model, sleep propensity at any given time integrates two independent biological mechanisms: a circadian regulator (process C) and a homeostatic regulator (process S). The circadian regulator is under control of the circadian clock and independent of external factors. The homeostatic regulator, on the other hand, is a tracker of past sleep and is responsible for the accumulation of sleep pressure upon sleep deprivation, or its release after a nap (*Borbély and Achermann, 1999*). The idea of a 'process S' is historically based upon electrophysiological recordings obtained in mammals and, in particular, on the observation that low-frequency electrophysiological correlates of neuronal activity — Slow Wave Sleep — increase with sleep deprivation, dissipate with sleep, and thus can ultimately act as biological markers for sleep pressure. The basic separation between a circadian and a homeostatic regulator, however, is a fundamental concept that lives beyond electrophysiology and can be adopted to model sleep pressure also in animals where the electroencephalographic (EEG) correlates of sleep are very different or unknown (*Campbell and Tobler, 1984*). In virtually all animals tested so far, sleep deprivation leads to a subsequent increase in sleep rebound (*Cirelli and Tononi, 2008*). Understanding the biological underpinnings of process C and process S is an important investigative task, not only to uncover the mechanisms regulating sleep, but ultimately its function too. Discovering how and why sleep pressure increases upon sleep deprivation may be critical to ultimately unravel what sleep is for.

Besides a homeostatic and a circadian controller, we do know that other factors can modulate sleep. Most people in western society will lament poor sleep habits and this is generally not due to malfunctioning of process S or process C but, instead, to societal or generally environmental and emotional causes (e.g. stress, anxiety, excitement, hunger, love) (*Ohayon, 2002*; *Adolescent Sleep Working Group et al., 2014*).

From the experimental perspective, changes in environmental temperature and food restriction constitute two important examples of sleep modulation by environmental conditions. In flies, an increase in temperature during the night has been shown to have profound effects on sleep pattern, but not necessarily on total sleep amounts (*Lamaze et al., 2017*; *Parisky et al., 2016*). In rats (*Danguir and Nicolaidis, 1979*), humans (*MacFadyen et al., 1973*), and flies (*Keene et al., 2010*)

*For correspondence:
giorgio@gilest.ro

Competing interests: The authors declare that no competing interests exist.

**eLife digest** Humans spend one-third of their lifetime sleeping, but why we (and other animals) need to sleep remains an unresolved mystery of biology. Our desire to sleep changes depending on how much sleep we've already had. If we've had a long nap during the day, we may find it harder to fall asleep at night; conversely, if we stay up all night partying, we'll have a difficult time staying alert the next day. This change in the pressure to sleep is known as "sleep homeostasis".

Can sleep homeostasis be suppressed? We know that some migratory birds are able to resist sleep while flying over the ocean. In addition, males of an Arctic bird species forgo sleep for courtship during the three-week window every year when females of its species are fertile. These examples suggest that some behavioral or environmental factors may influence sleep homeostasis.

Beckwith et al. now show that sexual arousal can disrupt sleep homeostasis in fruit flies. In "blind date" experiments, young male fruit flies were kept in a small tube with female fruit flies, prompting a 24-hour period of courtship and mating. The males went without sleep during that period, and they did not make up for the lost sleep afterward.

In other experiments, male fruit flies were kept awake by a robot that disturbed them every time they tried to sleep. After such treatment, the flies normally attempted to nap. But if the sleep-deprived flies were exposed to a chemical emitted by female flies that increased their sexual arousal, they no longer needed to sleep.

Overall, the results presented by Beckwith et al. show that sleep is a biological drive that can be overcome under certain conditions. This will be important for sleep researchers to remember, because it means that it's possible to affect sleep regulation (perhaps by making the animal stressed or aroused) without activating the brain circuits directly involved in regulating sleep.
DOI: https://doi.org/10.7554/eLife.27445.002

starvation has been shown to lead to a rapid decrease in sleep amount. In mammals, this also correlates with qualitative differences in the EEG pattern (*Danguir and Nicolaidis, 1979*). Besides a strong evolutionary conservation at the behavioural level, caloric intake and sleep are also genetically linked, as the same proteins and neuromodulators have been shown to control both (*Willie et al., 2001*). However, the relationship between the two is also complicated by the fact that caloric restriction has profound consequences on metabolism.

Here we describe a new paradigm to study the behavioural, neuronal, and genetic connection between environment and sleep: sex drive. We find that, in male flies, sexual arousal has profound effects on sleep, and that sexual experience or even exposure to pheromones alone are sufficient stimuli to counteract sleep pressure after sleep deprivation.

## Results

### Paradoxical effects of social sleep deprivation on sleep rebound

After being forcefully deprived of sleep, *Drosophila melanogaster* consistently show an increase in sleep pressure, in the form of a concomitant increase in sleep amount (*Huber et al., 2004*) and in arousal threshold (*Faville et al., 2015*). In other words, flies, like mammals, appear to sleep longer and deeper after sleep deprivation and both are clear signs of what is normally referred as 'sleep rebound', a hallmark of sleep homoeostasis. To deprive flies of sleep, most researchers would use mechanical machines, such as laboratory shakers, that subject animals to frequent, if not continuous, vibratory stimuli (*Faville et al., 2015*; *Huber et al., 2004*). We previously showed that a spatially restricted, forced interaction between two males also leads to a robust sleep deprivation that has the same behavioural and cellular characteristics of mechanical deprivation, including a similar extent of detectable rebound and comparable biochemical correlates (*Gilestro et al., 2009*). To further investigate how social interaction affects sleep, we devised an experimental paradigm based on computer-assisted video analysis of behaviour. Using *ethoscopes*, video tracking machines recently developed in our laboratory (*Geissmann et al., 2017*), we monitored and annotated the behaviour of flies either in isolation (baseline and rebound days) or in groups of two (interaction day). The advantages of using video tracking over the infrared beam split system when measuring sleep have

been discussed at length elsewhere (*Donelson et al., 2012*; *Gilestro, 2012*; *Zimmerman et al., 2008*) and, arguably, video tracking becomes even more compelling when monitoring multiple flies interacting in the same space.

In our archetypical experiment, wild-type male flies (CantonS) were kept in social isolation in small glass tubes for five days, in order to acclimatise to their environment and to record their baseline activity (see Materials and methods). Then, at the beginning of interaction day, we introduced a second individual in the restricted recording space: the *intruder*. For MM interactions, the intruder was another male of different eye colours (*white*[1118], cyan in figures). For MF interactions, a wild-type virgin female (peach in figures). Mock control male animals underwent the same experimental manipulation but were kept in isolation also during interaction day (mock, grey in figures). In all cases, interaction lasted no more than 24 hr. Confirming previous results (*Gilestro et al., 2009*), we found that MM interactions consistently led to a sleep deprivation during the interaction period, and to a noticeable rebound immediately after (*Figure 1A–D* and *Figure 1—figure supplement 1A,B*). The MF interaction led to an even greater deprivation of sleep (*Figure 1C*) but, surprisingly, did not show any subsequent rebound (peach in *Figure 1B*). Why?

## Social interaction leads to quantitatively and qualitatively different sleep deprivations

One first explanation could be that our tracking system overestimates the extent of sleep deprivation experienced in the MF interaction. To explore this possibility, we video-recorded interacting animals and manually scored their behaviour (*Figure 1D* and interactive video currently available at https://lab.gilest.ro/projects/raw-data/regulation-of-sleep-homeostasis-by-sex-pheromones-supplementary-videos/ – MF: 1082 bins scored per day; MM: 1247 bins scored per day) as well as their euclidean coordinates (scored 347 times in a day for Mock, MM and MF. $N_{mock}$ = 12, $N_{MM}$ = 11, $N_{MF}$ = 11). Human scoring confirmed machine scoring, as well as previous results (*Fujii et al., 2007*), and showed that the MF interaction led indeed to a sustained increase in activity (*Figure 1D* and *Figure 1—figure supplement 2*). In particular, even though all couples copulated within minutes from the start of the interaction (16.6 ± 15.7 min; mean ±SD), male flies still spent on average 47% of their time actively courting the female (47 ± 16% over 24 hr; 61 ± 25% during the day and 33 ± 12% during the night; mean ±SD). Flies engaged in MM interaction, on the other hand, were not as physically active as flies in MF (*Figure 1D* and *Figure 1—figure supplement 2*), thus not explaining but instead reinforcing the apparent paradox of absence of sleep rebound after interaction with a female.

To further characterise the consequences of social interaction, we also used a recently established CaLexA assay (*Liu et al., 2016*) to compare, *a posteriori,* the neuronal activity in the R2 neurons of the ellipsoid body after 24 hr of social interaction (MM or MF) or 24 hr of mechanical sleep deprivation (*Figure 1E,F*). The CaLexA system uses a calcium-responsive transcription factor to drive a green fluorescent protein (GFP) in neurons that undergo prolonged firing activity (*Masuyama et al., 2012*). Firing rate of R2 neurons was shown to correlate with sleep drive, therefore an increase of CaLexA fluorescence in those neurons can be interpreted as a *bona fide* proxy for neuronal firing, and ultimately, for sleep pressure (*Liu et al., 2016*). In all three experimental conditions, R2 neurons labelled by the R30G03-GAL4 driver showed a sustained and similar increase in detectable CaLexA-GFP levels compared to mock (*Figure 1E,F*), suggesting that all three conditions elicit a comparably efficient sleep deprivation.

## Rebound sleep is regulated by species-specific pheromones

Together, these results show that males who engaged in sexual interaction (a) experience a highly efficient sleep deprivation (*Figure 1C* and *Figure 1—figure supplement 2*) and (b) exhibit increases in neuronal markers which typically appear after prolonged wakefulness (*Figure 1E and F*). Why do they show no rebound sleep, then? One possibility is that the memory of their recent sexual encounter could motivate them to keep searching for a mating partner. To test this hypothesis, we subjected two canonical memory mutants to the same experimental paradigm: *dunce* and *rutabaga* (*Figure 2A,B* and *Figure 2—figure supplement 1A,B*). Both mutants are amongst the first and the best-characterised memory mutants discovered in *Drosophila* (*Davis and Dauwalder, 1991*; *Levin et al., 1992*) and have been shown to be unable to consolidate memory in many paradigmatic

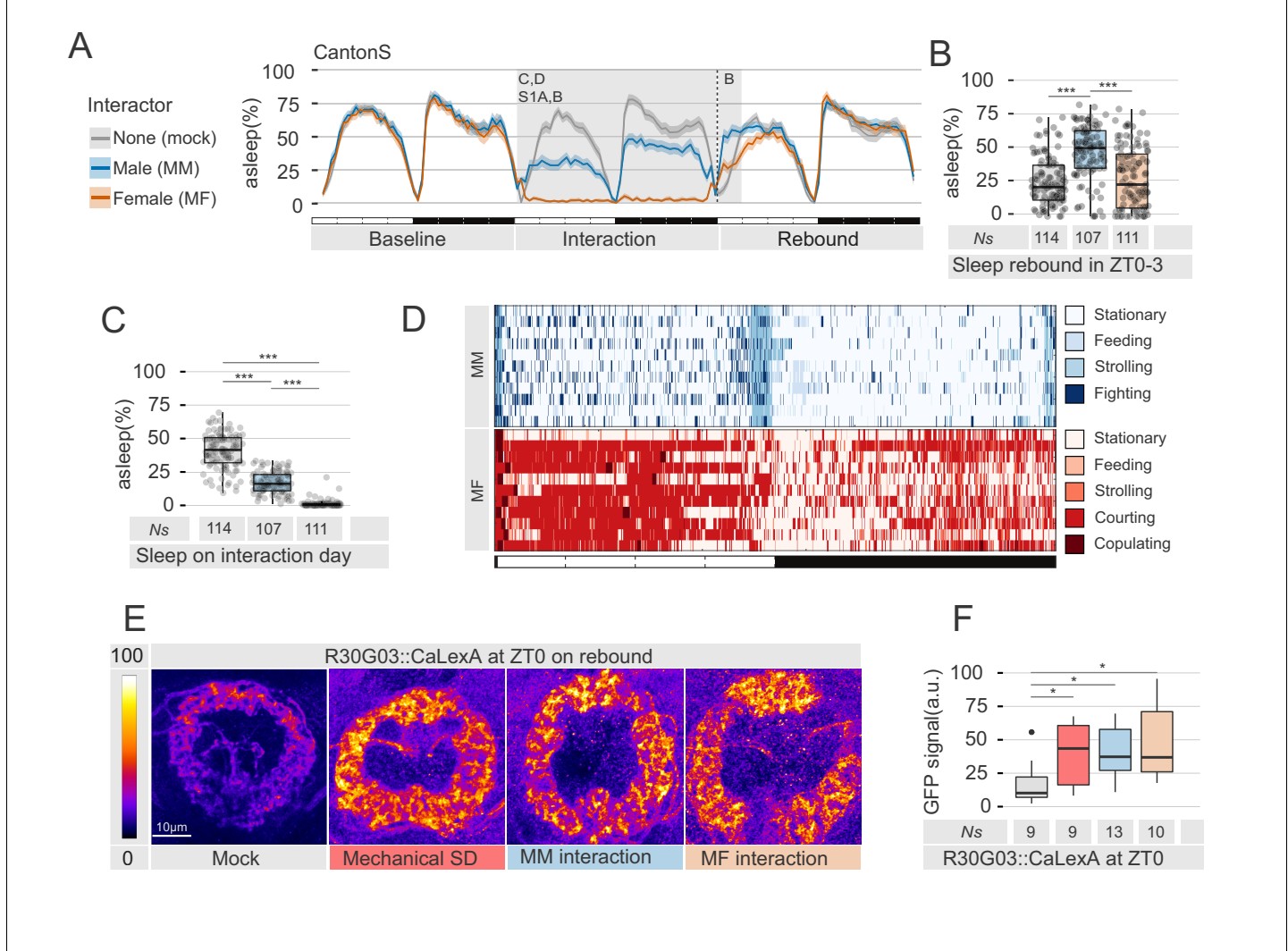

**Figure 1.** Social interaction leads to paradoxical effects on sleep rebound. (**A**) Sleep profile of socially isolated male CantonS flies that were forced to interact for 24 hr with wild-type virgin females (MF, peach), white eyed males (MM, cyan) or sham manipulated (mock, grey). The shaded grey background indicate the area analysed in B, C and D. The vertical dash line indicates the time point for brain dissections shown in E,F. (**B**) Quantification of sleep rebound during ZT0-3 on rebound day for the experiments shown in A. Ns under the bar plots indicate the number of animals used. (**C**) Quantification of sleep amount on the interaction day for flies shown in A. On interaction day, mock were estimated to be asleep 40.6 ± 13% of the time; MM 16 ± 8%; MF 1.1 ± 2% – mean ±SD. (**D**) Representative behavioural classification of the interaction day for MM coupling (upper panel n = 11) or MF coupling (lower panel n = 11). Each row shows the behavioural profile of a male during 24 hr of social interaction. (**E**) Representative image of whole-mount anti-GFP immunostained flies expressing CaLexA in the ellipsoid bodies using the R30G03 driver. Scale bar 10 μm. (**F**) Quantification of the experiment in E. In all figures, * indicates a p<0.05; **p<0.01; ***p<0.001 – pairwise Wilcoxon rank sum test with Benjamini and Hochberg correction. In all ethograms, the dark coloured lines indicate the mean values for sleep while the opaque borders indicate 95% bootstrap resampling confidence interval (see Materials and methods).

DOI: https://doi.org/10.7554/eLife.27445.003

The following figure supplements are available for figure 1:

**Figure supplement 1.** Analysis of sleep rebound.
DOI: https://doi.org/10.7554/eLife.27445.004

**Figure supplement 2.** Human assisted behavioural scoring of single fly sleep in Mock, solitary condition (**A**), during MF interaction (**B**), or MM interaction (**C**) to validate automatic scoring.
DOI: https://doi.org/10.7554/eLife.27445.005

**Figure supplement 3.** DAM data of the main archetypical phenotype of this work, published for sake of reproducibility.
DOI: https://doi.org/10.7554/eLife.27445.006

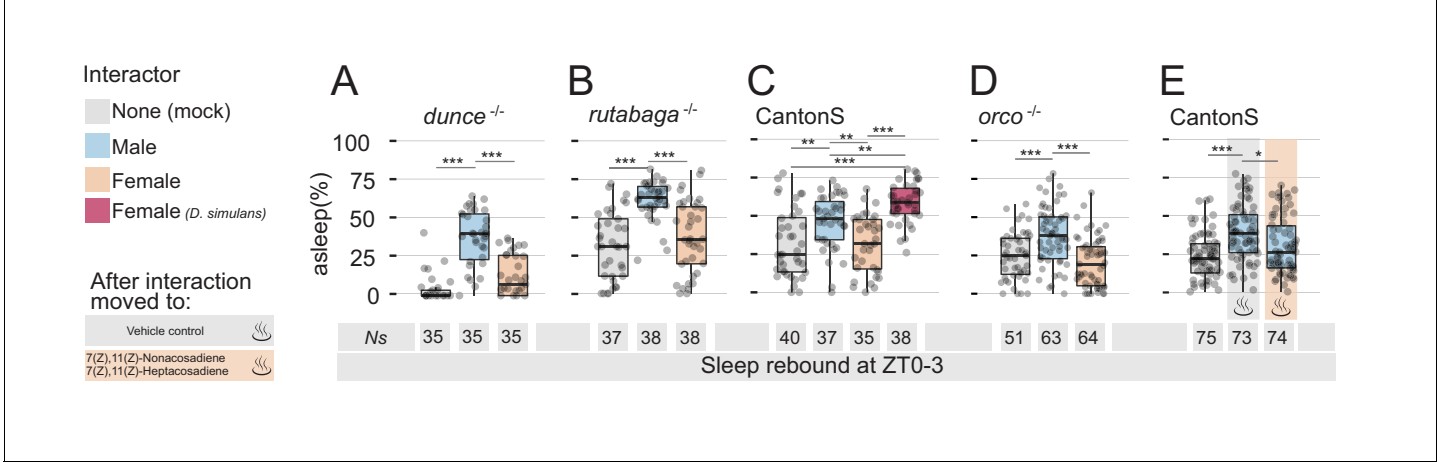

**Figure 2.** Detection of non-volatile female pheromones is sufficient to suppress sleep rebound after sleep deprivation. All graphs show quantification of sleep rebound during ZT0-3 on rebound day. Total number of animals is shown under the box plots. Rebound sleep in the memory defective *dunce*[1] (**A**) and in *rutabaga*[1] (**B**) mutants after Mock, MM, or MF. Legend on the left of (**A**) applies to the entire figure. (**C**) CantonS male flies after 24 hr of interaction with other males, *D. melanogaster* females, *D. simulans* females or in mock control. (**D**) Rebound sleep in anosmic *orco*[1] mutants after Mock interaction, MM, or MF. (**E**) CantonS male flies after mock interaction or interaction with white eyed males followed, at rebound, by vehicle control or by female pheromones.

DOI: https://doi.org/10.7554/eLife.27445.007

The following figure supplements are available for figure 2:

**Figure supplement 1.** Three day sleep ethograms of the experiments described in *Figure 2*.

DOI: https://doi.org/10.7554/eLife.27445.008

**Figure supplement 2.** Inter-specific male-female interaction.

DOI: https://doi.org/10.7554/eLife.27445.009

conditions, including courtship conditioning (*Griffith and Ejima, 2009*). While the role of *dunce* and *rutabaga* in the context of courtship conditioning is well described (*Joiner and Griffith, 1999*), it is not known whether flies possess any memory of past sexual experience. We speculated that if recollection of past experience is responsible for the suppression of rebound, one may expect to see a regular rebound in forgetful flies. This was not the case (*Figure 2A,B* and *Figure 2—figure supplement 1A,B*). As observed in wild-type flies, learning mutants also experienced a strong degree of sleep deprivation when forced to interact with females (*Figure 2—figure supplement 1A,B*), but they also lacked sleep rebound the day after. Regular rebound was once again observed after MM interaction.

If it is not a memory of the past experience that is responsible for the suppression of sleep rebound, could it be due to a physical trace left in the environment? Could either a volatile or non-volatile sex pheromone be left in the tube after the MF interaction, thus contributing to prolonging a signal of sexual arousal? We reckoned one way to approach this hypothesis would be to force an inter-species sexual interaction: a large number of olfactory and gustatory stimuli contributes to the complex courtship ritual between males and females (*Dweck et al., 2015*) and a convenient way to rule many at once is to force interaction between *D. melanogaster* males and a close evolutionary relative, such as *D. simulans* (*Manning, 1959*; *Sturtevant, 1919*). We therefore placed *D. melanogaster* wild-type males with *D. simulans* females on interaction day and video-recorded, then scored, their behaviour. In accordance with the classical literature (*Schilcher and Dow, 1977*), the inter-species MF interaction resulted in limited copulation (only 2 flies out of the 11 that were visually monitored, *Figure 2—figure supplement 2A*), but with some degree of courting mainly during the day (10.8 ± 2.3% over 24 hr but only 2.3 ± 1.4% during ZT 12–24; mean ±SD), followed by sleep deprivation throughout the night (*Figure 2—figure supplement 2B*). However, after inter-species MF interaction, male flies finally did show a sleep rebound that was even greater than the rebound observed after MM interactions (*Figure 2C* and *Figure 2—figure supplement 1C*). Interestingly, even though we never observed fighting behaviour between *D. melanogaster* males and *D. simulans*

females, the activity profile, the limited courtship, and the rebound were more reminiscent of MM interaction than MF interaction.

## Role of non-volatile pheromones

The sleep rebound observed after inter-species interaction suggests that a possibly arousing chemical signal left by the female may be responsible for the suppression of rebound after *D. melanogaster* specific MF interaction. In *Drosophila*, some pheromones have a certain degree of volatility (*Farine et al., 2012*) and, to test whether an olfactory signal was involved with this process, we measured rebound after social interaction in the anosmic *orco* mutants (*Larsson et al., 2004*) but found no difference between wild-type and *orco* mutant flies: anosmic males also lacked rebound after MF-induced sleep deprivation (*Figure 2D* and *Figure 2—figure supplement 1D*). If there is an arousing signal that anosmic flies can still perceive, could this be a non-volatile pheromone? Females of *D. melanogaster* and *D. simulans* have different cuticular hydrocarbons acting as sex pheromones, with the former bearing predominantly 7,11-Heptacosadiene (7,11-ND) and the latter 7-Tricosene (7-HD) (*Jallon, 1984*; *Marcillac et al., 2005*). If olfactory signals are not involved, we reasoned that a *D. melanogaster* specific cuticle pheromone could be responsible for the puzzling phenotype. We, therefore, subjected wild-type male flies to MM interaction and then, at the dawn of rebound day, we removed the intruder and inserted in the recording tube a fragment of paper on which we had previously diluted a mix of the species-specific sex pheromones 7,11-ND and 7,11-HD or the solvent alone as control (*Figure 2E* and *Figure 2—figure supplement 1E*). At last, we found that the mere presence of *D. melanogaster* female cuticular pheromones could indeed inhibit sleep rebound after MM interaction, suggesting the pheromones left by the female were sufficient to counteract sleep pressure accumulated at rebound day.

Male flies sense female non-volatile pheromones through neurons located on the distal tip of their forelegs (*Lu et al., 2012*; *Starostina et al., 2012*; *Thistle et al., 2012*; *Toda et al., 2012*; *Vijayan et al., 2014*). At the beginning of the sexual courtship ritual, male flies tap the female to presumably taste and recognise sex-specific signals (*Spieth, 1974*) that are important for courting to continue. In particular, 7,11-ND and 7,11-HD are sensed by neurons expressing members of the degenerin/epithelial sodium channel (DEG/ENaC) family — Ppk25, Ppk23, and Ppk29 — and male flies mutants in either of these receptors show a decreased level of courtship (*Liu et al., 2012*; *Starostina et al., 2012*; *Toda et al., 2012*; *Vijayan et al., 2014*). Behavioural and electrophysiological data showed that sex-pheromones detection is almost completely lost in *ppk23* mutant males (*Lu et al., 2012*; *Thistle et al., 2012*; *Toda et al., 2012*) and therefore we reasoned that *ppk23* mutants could serve as a good model to test, once more, the hypothesis that suppression of sleep rebound is due to pheromone signalling. We then subjected male *ppk23* mutant flies to four experimental conditions: MF interaction, MM interaction, and MM interaction with or without the addition of exogenous pheromones (*Figure 3*).

As predicted, flies underwent the expected level of sleep deprivations (*Figure 3B–E*), but MF condition did not show an abnormal sleep rebound (*Figure 3A*), indicating that *ppk23* signalling during the interaction plays a role in suppressing sleep rebound.

## Pheromone stimulation is sufficient to suppress sleep rebound after sleep deprivation

The data collected until this point show that male flies, exposed to female pheromone, will downregulate their natural need for sleep rebound after sleep deprivation. However, the attentive reader will have realised that all experiments performed so far rely on a social paradigm for sleep deprivation, therefore introducing a confounding condition: is the mere presence of pheromones truly sufficient to suppress sleep deprivation, or is this somehow connected to the social nature of our behavioural paradigm? After all, we do know that social interaction in flies can have profound effects on their sleep (*Ganguly-Fitzgerald et al., 2006*). To test sufficiency of pheromones effect on sleep rebound, we devised two sets of experiments, in which we replaced social-driven sleep deprivation with mechanical sleep deprivation, using the sleep deprivation module of our ethoscopes (*Figures 4* and *5*). Ethoscopes can interact with single flies in a context dependant manner, triggering events upon a real-time analysis of behaviour (*Geissmann et al., 2017*). We programmed the ethoscopes to rotate a tube whenever the animal inside was detected completely inactive for 60 s. We call this

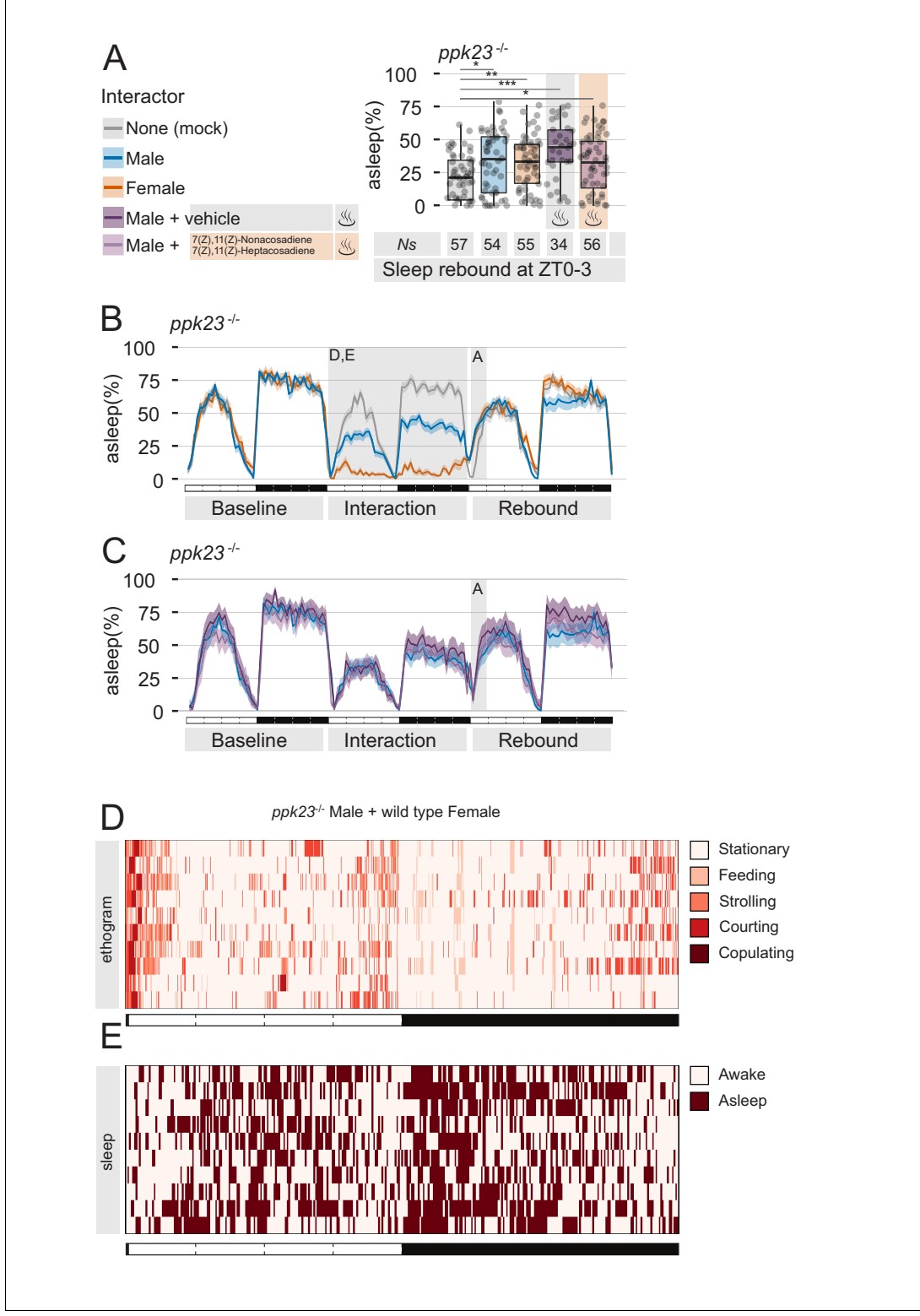

**Figure 3.** The pheromone receptor Ppk23 is necessary for sleep suppression after sleep deprivation. (**A**) Rebound during ZT0-3 in *ppk23*$^\Delta$ mutants after Mock interaction, MM, MF, MM followed by control vehicle, or MM followed by female pheromones. Legend on the left applies to A,B,C. (**B,C**) Three day sleep ethograms of the experimental conditions shown in (**A**). The data are split in two panels for sake of visualisation. The same MM line (blue) is reproduced in both. (**D**) Representative behavioural classification obtained with human scoring in MF interaction

*Figure 3 continued on next page*

Figure 3 continued

between *ppk23*^Δ mutant males and wild type females. Compare with wild-type MF in **Figure 1D**. (**E**) Human scoring of single fly sleep for a subset (N = 10) of the flies shown in C.

DOI: https://doi.org/10.7554/eLife.27445.010

paradigm: dynamic sleep deprivation (*Geissmann et al., 2017*). In the first set of experiments (*Figure 4*), CantonS flies were subjected to dynamic SD for 24 hr, then transferred into their same tube (mock manipulation – *Figure 4A,B*), to a clean and fresh tube (*Figure 4C,D*), or transferred into a tube where a virgin female was previously housed for five days (*Figure 4E,F*). Sleep rebound after SD was observed in the first two cases, but not in the last, suggesting that pheromones left by the previously hosted female are indeed sufficient to suppress rebound. To ultimately test the sufficiency of sex pheromones, we conducted a second set of experiments in which flies were engineered to express the thermo-activated channel TrpA1 in the pheromone sensing cells expressing the *ppk23*-GAL4 driver (*Figure 5*). These flies were also subjected to dynamic SD for 24 hr and then the temperature was raised from the inactivating (22°C) to the activating (29°C) condition to synaesthetically stimulate pheromone sensation. Of all experimental conditions (*Figure 5* and *Figure 5—figure supplement 1*), lack of rebound after SD was observed only when pheromone sensing cells were thermogenetically stimulated (*Figure 5E,F*).

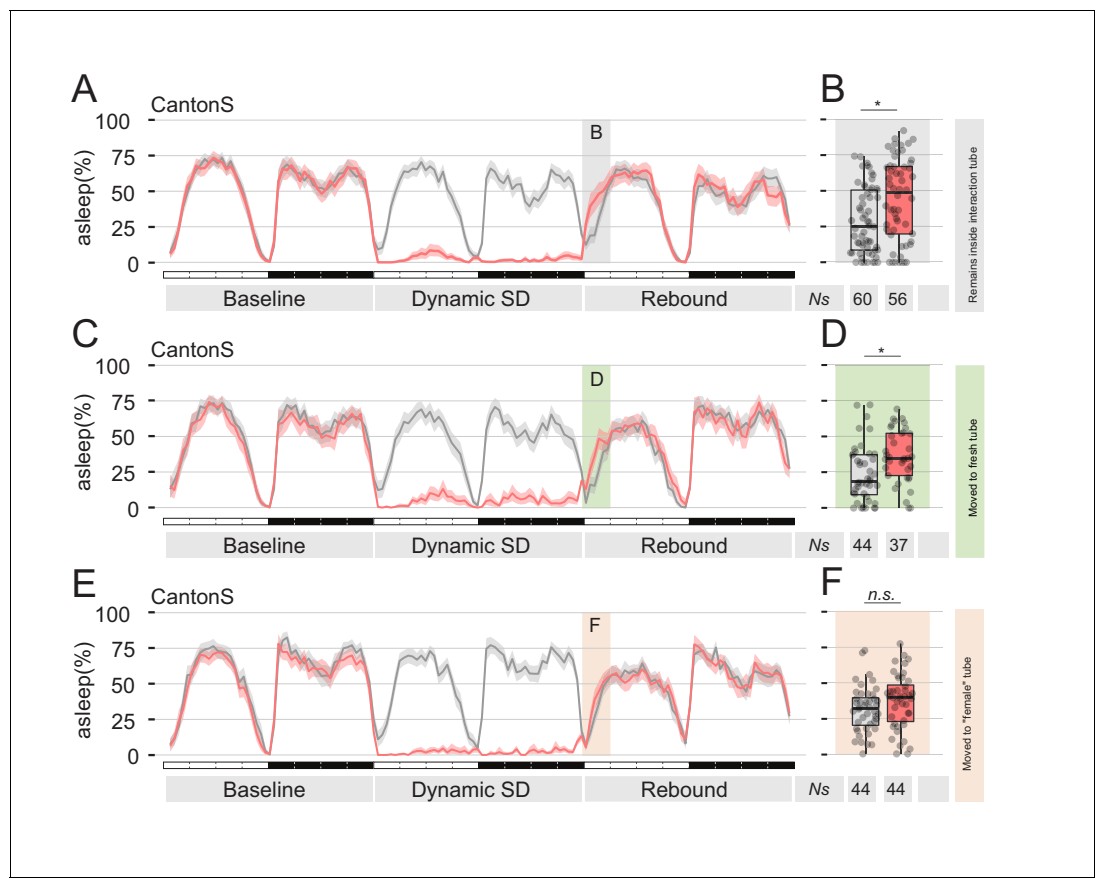

**Figure 4.** Presence of female pheromones is sufficient to suppress sleep rebound after sleep deprivation. (**A,C,E**) Sleep profile of Cantons S flies that, immediately after sleep deprivation, were sham transferred into the same tube (**A**), into a clean fresh tube (**C**), or into an empty tube were a virgin female fly was previously housed for 24 hr (**E**, 'female tube'). In all panels, grey lines show mock conditions that underwent the same treatment but were not sleep deprived. (**B, D and F**) Quantification of ZT0-3 rebound for A, C, and E respectively.

DOI: https://doi.org/10.7554/eLife.27445.011

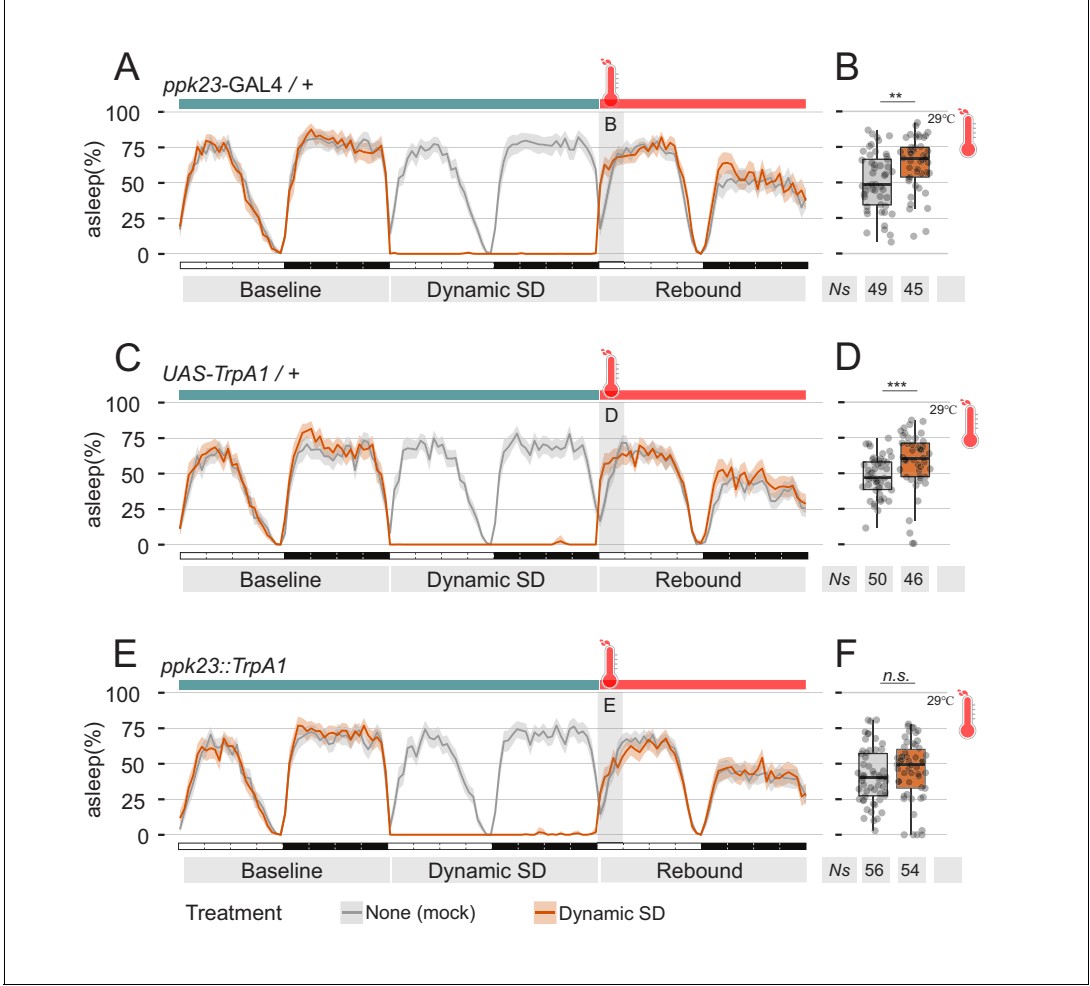

**Figure 5.** Manipulation of pheromones sensing cells is sufficient to suppress sleep. (A–D) Parental control genotypes (E,F) flies expressing the thermo-activated channel TrpA1 under control of *ppk23* GAL4. Red traces indicate the sleep pattern of flies subjected to mechanical sleep deprivation at the non-activating temperature (22°C); during rebound time ZT0-3, temperature was raised to 29°C. Grey lines indicate the sleep profile of mock control flies that underwent the same temperature treatment but were not sleep deprived. (B, D, F) Quantification of sleep rebound during ZT0-3 of rebound day for flies shown in A, C, and E, respectively.

DOI: https://doi.org/10.7554/eLife.27445.012

The following figure supplement is available for figure 5:

**Figure supplement 1.** Temperature control conditions for experiments shown in *Figure 5*.

DOI: https://doi.org/10.7554/eLife.27445.013

## General role of pheromones and sexual arousal in sleep control

The results collected so far indicate that pheromone signalling has the ability to suppress sleep rebound after sleep deprivation. However, is pheromone the only cue able to do so? After all, suppression of sleep observed in the archetypical experiment (*Figure 1A*) appears to be even stronger than the suppression observed after exposing flies to pheromone alone. To address this question, we performed two sets of experiments. In a first set, we subjected flies to the usual social interaction paradigm but, at the end of the interaction day, we removed the intruders and transferred the focal flies not in their own tube as done previously, but instead in a clean, fresh tube (*Figure 6A,B*). Indeed, we found that also when transferred to a clean tube — and therefore in the absence of residual female pheromones in their environment — male flies did show a suppressed rebound (*Figure 6A,B* - compare with *Figure 1A,B*). In a second set of experiments, we used the thermosensitive form of the neuronal inhibitor *shibire* (*shiTS*) to selectively silence *ppk23* neurons at the dawn of rebound day, after social interaction (*Figure 6C–H*). As expected, silencing of the pheromone

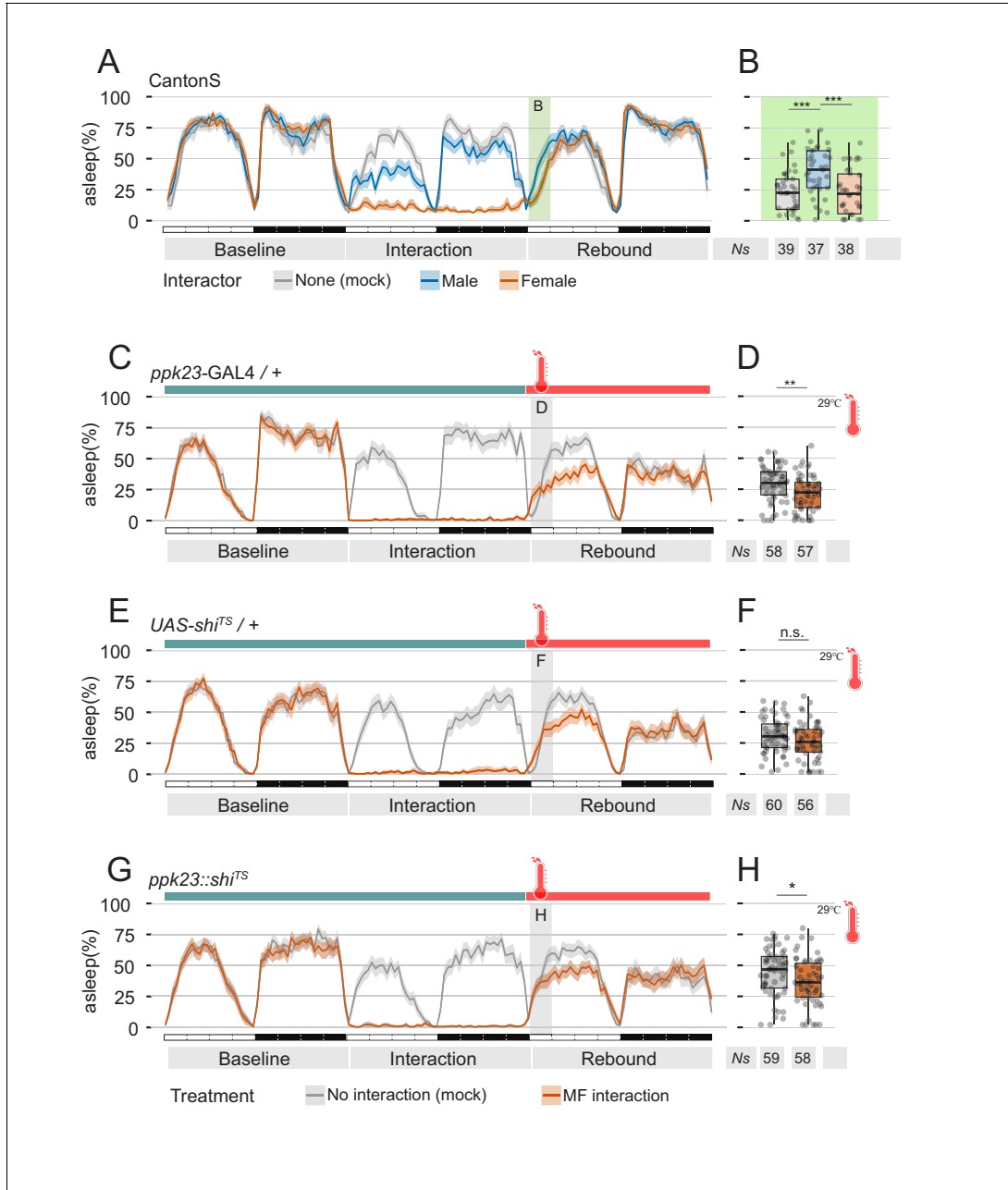

**Figure 6.** Perception of female pheromones after social interaction is not required to suppress sleep. (**A**) Three day sleep profile of CantonS flies, showing baseline day, interaction day and rebound day. At the end of interaction day, flies were mock transferred into a clean, fresh tube. (**B**) Quantification of rebound at ZT0-3 for the experiment shown in A. (**C–H**) Inactivation of *ppk23* cells during rebound does not rescue the sleep phenotype. Parental control genotypes (**C–E**) flies carrying either the *ppk23* GAL4 insertion (**C,D**) or the UAS-*shi*^TS^ insertion (**E,F**). Experimental flies carrying both (**G,H**). Red traces indicate the sleep pattern of flies subjected to MF interaction at non-activating temperatures (22°C); on rebound day, temperature was raised to 29°C. Grey lines indicate the sleep profile of mock control flies that underwent the same temperature treatment but did not experience social interaction. (**D, F, H**) Quantification of sleep rebound during ZT0-3 of rebound day for flies shown in C, E. and G, respectively.
DOI: https://doi.org/10.7554/eLife.27445.014

signalling cells after MF interaction did not rescue the sleep rebound phenotype. These findings strongly suggests that suppression of sleep rebound may be due to a general state of sexual arousal, which could be elicited either by the recent sexual experience or by the presence of sex pheromones: the combination of both factors may then act in synergy to manifest the stronger effect shown in *Figure 1A,B*.

Sexual arousal in *D. melanogaster* males is known to be largely under control of the sexually dimorphic P1 cluster of *fruitless* expressing neurons (*Yamamoto and Koganezawa, 2013*). P1 neurons are activated by contact with females (*Kohatsu and Yamamoto, 2015*) and, conversely, experimental activation of P1 neurons is sufficient to trigger or enhance courtship behaviour (*Kohatsu et al., 2011*), possibly generating an internal state of sexual arousal. Therefore, to test the ultimate shape of our hypothesis, we expressed the thermoactivated channel TrpA1 in the P1 neurons and raised the temperature to the activating condition (29°C) for 24 hr (*Figure 7*). Sustained and prolonged activation of P1 neurons led to a phenotype of prolonged activity and almost total suppression of sleep (*Figure 7D*), largely similar to the one observed when pairing a male fly with a female partner. Most importantly, the sleep deprivation induced by activating P1 neurons also did not lead to sleep rebound but, on the contrary, to a noticeable reduction of sleep on rebound day (*Figure 7D,E*).

## Manipulating the extent and nature of sexual arousal

To further confirm that suppression of sleep rebound is indeed controlled by sexual arousal, we performed two final genetic manipulations, both involving the octopamine and tyramine synthesis pathway (*Figure 8A*). The octopaminergic system is a key regulator of *Drosophila* behaviour, involved among others, with modulation of sexual activity (*Huang et al., 2016*; *Zhou et al., 2012*), male-male aggression (*Zhou et al., 2008*) and sleep (*Crocker and Sehgal, 2008*; *Crocker et al., 2010*; *Yang et al., 2015*). In the first set of experiments, we subjected flies mutant for the TβH enzyme to the social interaction paradigm. In *TβH* mutant flies, octopamine synthesis is impaired and this has

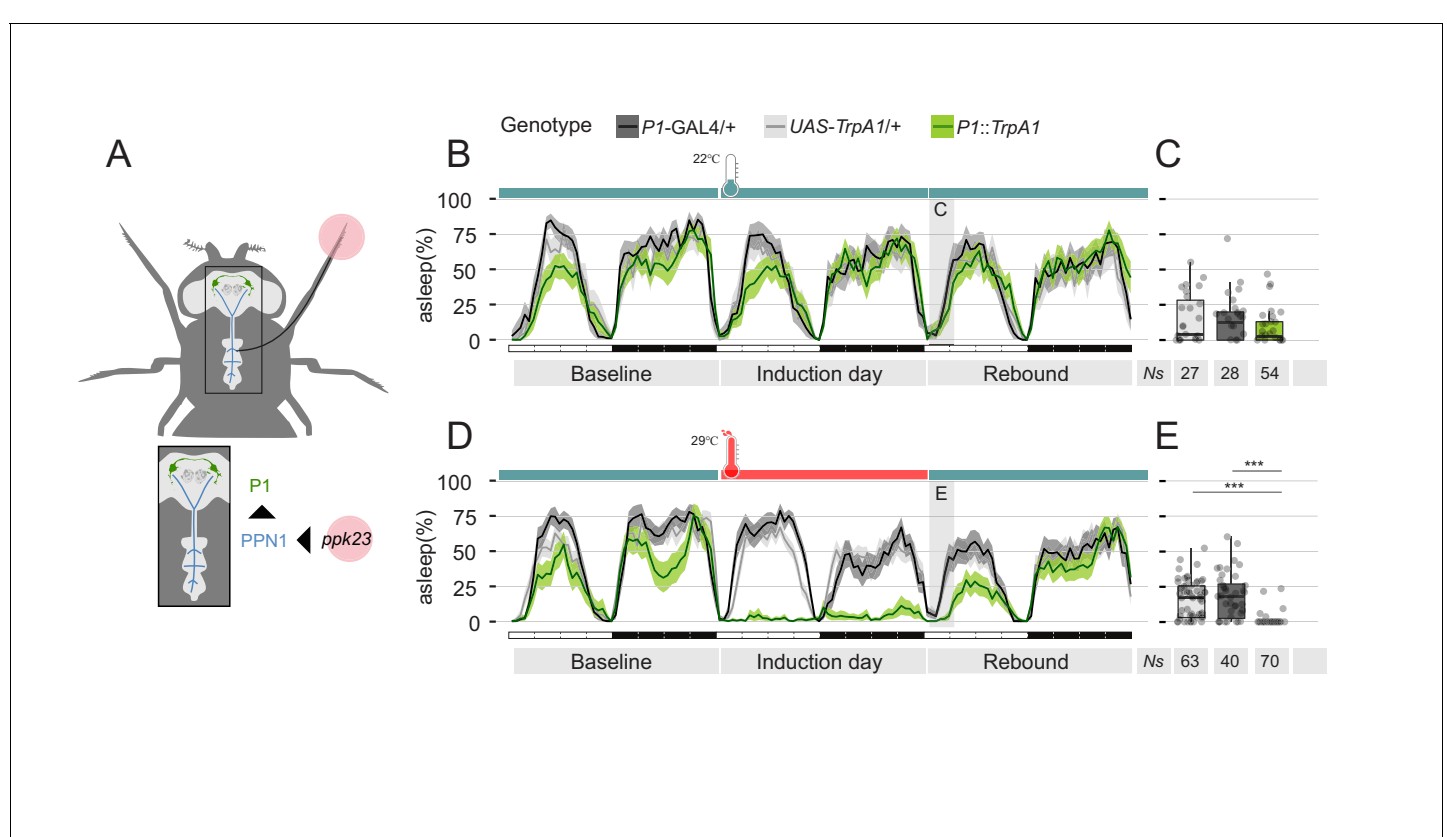

**Figure 7.** Residual sexual arousal after P1 activation leads to suppression of sleep rebound. (A) Diagram of the pheromone pathway. Signal from peripheral *ppk23* sensory neurons is transferred to PPN1 first and central P1 neurons ultimately. (B, D) Sleep profile of parental control lines (light and dark grey) or experimental line expressing the thermo-activated channel TrpA1 under control of the P1-split-GAL4 driver (green). Experimental flies (D) experienced a raise in temperature from 22°C to 29°C for 24 hr. On the following day, temperature was set again to 22°C. Control flies do not experience any temperature change. (C, E) Quantification of rebound at ZT0-3 for B and D respectively.
DOI: https://doi.org/10.7554/eLife.27445.015

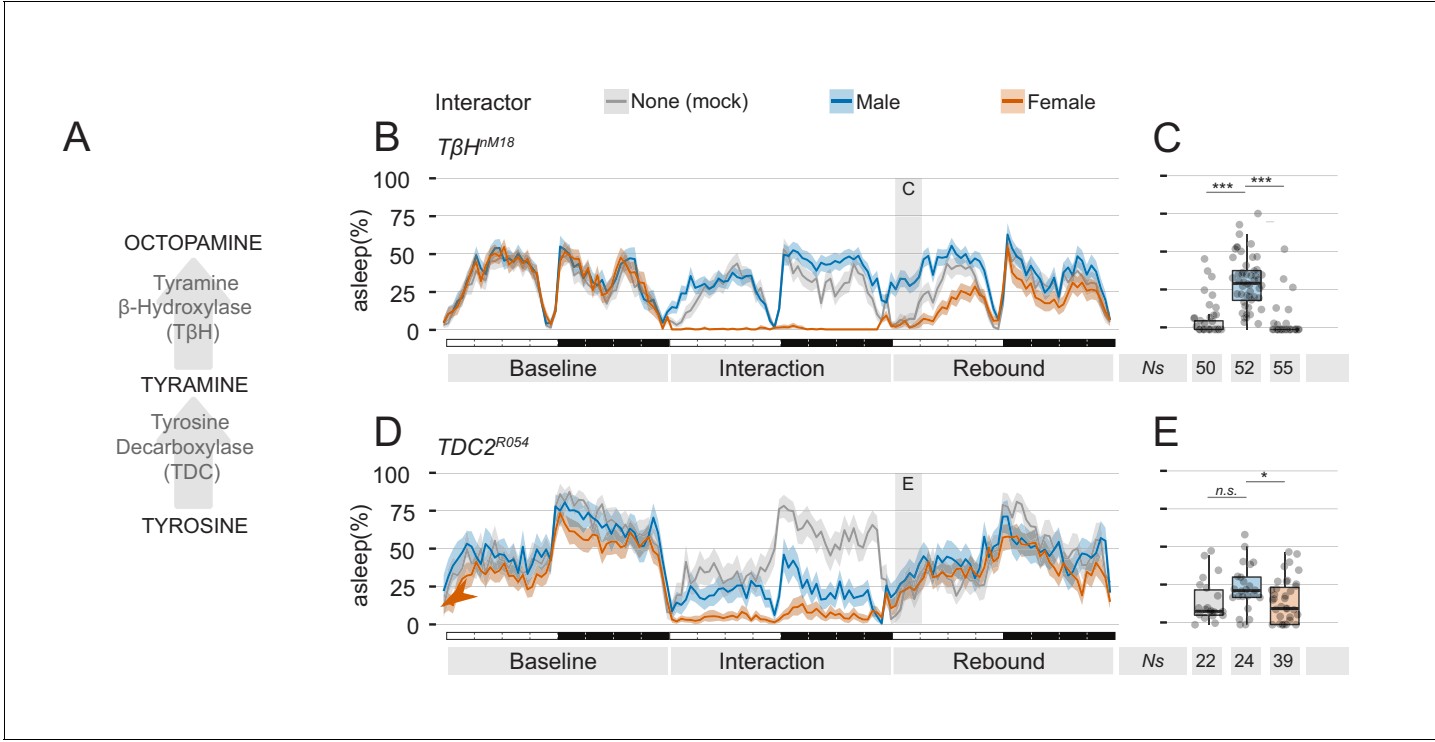

**Figure 8.** Mutants in the tyramine, octopamine pathway can be employed to manipulate quantity and quality of sexual arousal. (A) Diagram of the known pathway for generation of octopamine from tyrosine. (B) Three day ethogram plot for the sleep profile of TβH[nM18] mutant males subjected to social interaction. In these flies, the effect of MF interaction on sleep rebound is exacerbated. (C) Same experiment, using TDC2[R054] mutant male flies. In these flies, MM interaction also leads to suppression of sleep rebound. (C,E) Quantification of ZT0-3 for the experiments shown in B and D respectively.

DOI: https://doi.org/10.7554/eLife.27445.016

been linked to a deficit in the ability to form courtship conditioning (*Zhou et al., 2012*) a paradigm in which male flies learn to suppress their sexual instincts, after having been repeatedly rejected (*Griffith and Ejima, 2009*). In an educated guess, we reasoned that *TβH* mutant flies lacking courting conditioning may, therefore, show an abnormal increase of sexual arousal after prolonged social interaction and, possibly, show an increased effect in suppression of rebound. This was indeed the case (*Figure 8B,C*). *TβH* mutant flies showed a clear rebound after MM interaction and a strong suppression of rebound after MF interaction.

Finally, to investigate whether mere sexual arousal is responsible for this effect, we used flies mutant in the *TDC2* gene, that possess lower levels of tyramine and octopamine (*Crocker and Sehgal, 2008*) and were previously shown to court male as well as female flies (*Huang et al., 2016*). We hypothesised that if these flies are sexually aroused by both male and female partners, they should then respond with a suppression of sleep rebound to both conditions of social interaction. This was what we observed indeed (*Figure 8D,E*). In flies with a bi-sexual orientation, both MF and MM interaction lead to a strong suppression of sleep rebound.

## Discussion

The main finding of this work is that sexual arousal has the ability to modulate sleep pressure. We use different behavioural paradigms to promote a state of sexual arousal in male flies and show that, in all cases, this results in a suppression of sleep rebound following sleep deprivation. Why is this important?

In the past 15 years, *Drosophila* has emerged as one of the most promising animal models to study the biological underpinnings of sleep. Many genes that affect sleep in *Drosophila* have been identified so far, and many neuronal circuits that can alter sleep when manipulated have been

described (*Potdar and Sheeba, 2013*; *Tomita et al., 2017*). Given that the framework for sleep regulation is stably centred around the two-process model, newly identified neurons modulating sleep are normally classified either as involved with circadian regulation — and thus belonging to process C — or as involved with homeostatic regulation — and thus belonging to process S. Here, we identified an internal state that has the ability to modulate sleep and sleep pressure but arguably does not belong to either process.

Historically, accessory regulation of sleep has been attributed to neuromodulators and, again, *Drosophila* has proven instrumental in understanding how neuromodulators influence sleep (*Griffith, 2013*). However, environmental control of sleep is likely to extend beyond neuromodulators and indeed likely to encompass specific sensory and central circuits. Using optogenetics and thermogenetics, it is now possible to activate and silence single neurons or entire circuits looking for functional correlates of behaviour. A proper characterisation of possible outcomes is a necessary step: how can we distinguish if a neuron's main job is to *directly* regulate sleep pressure or, for instance, to create a state of anxiety, hunger or sexual arousal, that *indirectly* modulates sleep pressure? Paraphrasing a famous assay by Thomas Nagel, we cannot know *what is like to be a fly* (*Nagel, 1974*): does exposure to sex pheromones create an inner status of sexual arousal that then counteracts sleep, or does it directly interfere with sleep regulation without any further sexual implication? Manipulation of *ppk23* neurons, either thermogenetically or by the use of chemicals, does not elicit any clear sign of courtship (data not shown) and this is in accordance with previous literature, where it was also shown that activation of *ppk23* neurons alone is not sufficient to induce any sign of sexual behaviour in isolated flies (*Starostina et al., 2012*; *Toda et al., 2012*) and that the right pheromones can act, instead, to potentiate other concomitant sexual stimuli. In our paradigm, activation of P1 neurons also does not show any clear sign of courtship, such as singing through wing extension (data not shown).

For the sleep field, this work offers a novel experimental paradigm that could be used to dissect, in an ecologically meaningful way, how internal drives or environmental stimuli affect sleep regulation and sleep homeostasis. The interaction between sex and sleep in *Drosophila*, and more specifically the hierarchy of those two concurrent biological drives, was initially described in the frame of circadian interaction (*Fujii et al., 2007*) and very recently the neuronal underpinnings were investigated by two independent groups (*Chen et al., 2017*; *Machado et al., 2017*). In particular, *Machado et al. (2017)* and *Chen et al. (2017)* also find a role for the P1 neurons in diverting an animal's interest from sleep to sex. Our work, however, does not focus on the binary choice between sleep and courtship, but rather uncovers a new role for sexual arousal on modulation of sleep homeostasis, also in absence of a female partner. The concept that sleep homeostasis is not inviolable and can actually be modulated is not a novel one: migratory birds and cetaceans were reported to have the ability to suppress sleep at certain important periods of their lives, namely during migration or immediately after giving birth (*Fuchs et al., 2009*; *Lyamin et al., 2005*; *Rattenborg et al., 2004*); flies, similarly, were shown to lack sleep rebound after starvation-induced sleep deprivation (*Thimgan et al., 2010*) or after induction of sleep deprivation through specific neuronal clusters (*Seidner et al., 2015*). Perhaps even more fitting with our findings is the observation that male pectoral sandpipers, a type of Arctic bird, can forego sleep in favour of courtship during the three weeks time window of female fertility (*Lesku et al., 2012*). It appears, therefore, that animals are able to balance sleep needs with other, various, biological drives. It would be interesting to see whether these drives act to suppress sleep through a common regulatory circuit. Rebound sleep has always been considered one of the most important features of sleep itself. Together with the reported death by sleep deprivation, it is frequently used in support of the hypothesis that sleep is not an accessory phenomenon but a basic need of the organism (*Cirelli and Tononi, 2008*). Understanding the regulation of rebound sleep, therefore, may be crucial to understanding the very function of sleep. Interestingly, in our paradigm rebound sleep is not postponed, but rather eliminated. Moreover, on rebound day, the sleep architecture of sexually aroused male flies does not seem to be affected: the sleep bout numbers appear to be similar to their mock control counterparts, while the length of sleep bouts is, if anything, slightly reduced (*Figure 1—figure supplement 1*).

The last remark that arises from our finding concerns the use of *Drosophila melanogaster* as a model for complex brain functions, such as emotions. *Drosophila* neurobiology is experiencing a period of *renaissance*, driven by a Cambrian explosion of genomics, ethomics and connectomics. The field may soon be able to use fruit flies for behavioural models that were once considered to be

an exclusive of mammals - or even humans. Past examples of these behaviours are aggression or sleep itself. Studying emotions or internal states in animals is not an easy task, given their subjective nature. However, studying the effects of emotions on sleep may open a window of opportunity, by providing an easily quantifiable output.

## Materials and methods

### Fly stocks

Flies were raised under a 12 hr light:12 hr dark (LD) regimen at 25 on standard corn and yeast media. Following lines were used in the study: CantonS from Ralf Stanewsky (UCL, UK); *D. simulans* from Virginie Orgogozo (IJM, France); *ppk23*-GAL4 and *ppk23*$^\Delta$ mutants (*Toda et al., 2012*) from Barry Dickson (HHMI, USA); CaLexA (*Masuyama et al., 2012*) from Marc Dionne (ICL, UK); UAS-*shi*$^{TS}$ from James Jepson (UCL); R30G03-GAL4 (#49646) (*Liu et al., 2016*), *dunce*$^1$ (#6020), *rutabaga*$^1$ (#9404), and *orco*$^1$ (#23129) mutants, UAS-TrpA1 (#26263) from Bloomington Drosophila Stock Centre (Indiana, USA). The P1-split-GAL4 driver was created and provided by Eric Hoopfer (*Hoopfer et al., 2015*). TβH$^{nM18}$ and TDC2$^{R054}$ are from Stephen Goodwin (CNCB, Oxford).

### Neuronal activity in the R2 neurons of the ellipsoid body, CaLexA measurements

Animals were grown and treated in the same conditions as in behavioural experiments. After a day of social interaction or mechanical sleep deprivation, animals were anaesthetised and their brains were dissected and fixed as previously described (*Beckwith et al., 2013*). For CaLexA measurements, fly brains were immunostained with anti-GFP (1:400, ab290 Abcam). Images were taken under 40X magnification and analysed in Fiji/ImageJ (*Schindelin et al., 2012*). To measure signal intensities, a maximal intensity projection of all the stack comprising the R2 ring was generated. A doughnut shaped region of interest was superimposed to measure mean grey value for each R2 ring. Intensity on an adjacent non-labelled region was measured and subtracted. To allow comparison with previously published data (*Liu et al., 2016*), mechanical sleep deprivation was conducted by placing the flies on top of a laboratory shaker controlled by an Arduino timer activated in pulses of 5 to 30 s at pseudo-random intervals of 1 to 7 min (Arduino code and instructions on https://github.com/gilestrolab/fly-sleepdeprivator).

### Social interaction experiments

Sleep recordings were performed using *ethoscopes* (*Geissmann et al., 2017*) under 12:12 LD condition, 50–70% humidity, in incubators set at 25. In all experiments, environmental values of temperature, humidity, and light were recorded and monitored every 5 min. For social interactions, zero to one-day old flies were removed from a shared vial and placed in 70 mm x 5 mm glass tubes containing standard food. Twenty tubes were placed in each *ethoscope* arena. Flies were acclimated in behavioural glass tubes for 5 days of which the last 2 days were recorded as a baseline. On the interaction day, intruders (CantonS females or *white*$^{1118}$ males) were added at ZT0. Intruders were then removed from ZT23 to ZT24, finishing 10 min before the dark to light transition. Rebound period was then recorded for two consecutive days. All figures show the last baseline day and the first rebound day.

### Human scoring of social interaction

Manipulation of flies and recording of interaction was performed as in all other experiments with the only difference that experiments were video-recorded using the recording function of *ethoscopes*. Videos were recorded with a resolution of 1920 × 1080 pixels and a frame rate of 25 FPS. The degree of interaction was then scored using a web-based graphical interface, available upon request. Behavioural labelling was done at a frequency of approximately once every 60 s, while positional scoring with a frequency of once every 240 s.

### Pheromone delivery

For the pheromone experiments, a small fragment of 3 MM filter paper containing the pheromones mix (70 ng in 10 μl of 7(Z),11(Z)-Nonacosadiene and 70 ng in 10 μl of 7(Z),11(Z)-Heptacosadiene;

Cayman Chemicals, Ann Arbor, Michigan 48108 USA) or the vehicle (hexane) was added to the tube just after removal of the intruder male.

## Dynamic sleep deprivation

Sleep deprivation was conducted using the servo motors module of the *ethoscope* platform (*Geissmann et al., 2017*). All bouts of immobility lasting at least 60 s were automatically interrupted by the machine rotating individual experimental tubes, thus awakening the flies only when they were quiescent. For each stimulation, motors rotate three times: −85° 200 ms, +170° 300 ms, −85° 200 ms.

## Thermogenetics

For experiments employing *TrpA1* and *shi^TS^*, animals were raised in incubators set at 22°C. Baseline recordings and sleep deprivation were performed also in incubators set at the same temperature but the actual recorded temperature oscillated between 22°C and 24°C due to heat produced by *ethoscopes* themselves. Thermo-manipulation was conducted at 29°C. In all experiments, environmental conditions of light, temperature, and humidity were recorded with a frequency of once every 5 min. For the shiTS experiments shown in *Figure 6C–H*, the temperature was raised to 29°C at ZT23:30.

## Statistical analysis and data reproducibility

All data analysis was performed in R (*Core Team, 2014*) or in Python (*Team, 2015*). Behavioural data were analysed with the R package Rethomics (https://github.com/gilestrolab/rethomics) and statistical analysis consisted of pairwise Wilcoxon rank sum test (i.e. Mann–Whitney U test) with P value adjustment for multiple comparisons (*Benjamini and Hochberg, 1995*). For ethograms, bootstrap re-sampling with 5000 replicates, was performed in order to generate 95% confidence interval (*Carpenter and Bithell, 2000*) (shadowed ribbons around the mean in the figures). All experiments were replicated two to five times. In all figures, Ns represent the total number of flies over all experiments. Statistics were done on aggregated data. Outliers were never excluded. Flies that died during the course of the experiment were excluded from all analysis. All figures were generated in R, using ggplot2 (*Wickam, 2009*). For all boxplots, the bottom and top of the box (hinges) show the first and third quartiles, respectively. The horizontal line inside the box is the second quartile (median). Tuckey's rule (the default), was used to draw the 'whiskers' (vertical lines): the whiskers extend to last extreme values within ±1.5 IQR, from the hinges, where IQR is Q3-Q1. A detail summary of all statistical comparisons is provided as *Supplementary file 1*.

## Supplementary videos

Supplementary Videos S1, S2, S3, S4 (available at https://lab.gilest.ro/projects/raw-data/regulation-of-sleep-homeostasis-by-sex-pheromones-supplementary-videos/ given the interactive nature of this figure). (S1) Interaction between wild-type *D. melanogaster* male and female flies (MF). (S2) Interaction between wild-type *D. melanogaster* male and white eyed males (MM). (S3) Interaction between *ppk23^Δ^* mutant *D. melanogaster* male and wild-type *D. melanogaster* female flies. (S4) Interaction between wild-type males and *D. simulans* female. S1 and S2 show the same dataset as in *Figure 1D*. S3 has the same dataset as in *Figure 3D*. S4 shows the same dataset as in *Figure 2—figure supplement 2*. In all videos, hover the mouse cursor on the ethogram to highlight the corresponding region of interest in the video. Click on the ethogram or on the day/night bar to seek to the relative video position. Legend for behavioural classification as in *Figure 1C*. Videos are compressed to facilitate access. Full, uncompressed dataset accessible on Zenodo via doi: 10.5281/zenodo.167551 or https://zenodo.org/record/167551#.Wbo5OkqGMsk (*Gilestro et al., 2016*). When clicking on ethograms, allow an error on time axis of ±10 min on the relative video.

## Acknowledgements

The research leading to these results has received funding from BBSRC through BB/M003930/1, BB/J014575/1, and 'Imperial College London – BBSRC Impact Acceleration Account'. EJB was supported by EMBO ALTF 57–2014 and by the People Programme (Marie Curie Actions) of the

European Union's Eighth Framework Programme H2020 under REA grant agreement 705930. We thank the Imperial College London Advanced Hackspace (ICAH) and the the Facility for Imaging by Light Microscopy (FILM) at Imperial College London, which is part-supported by funding from the Wellcome Trust (grant 104931/Z/14/Z) and BBSRC (grant BB/L015129/1). We would like to thank Ralf Stanewsky, Virginie Orgogozo, Claudio W Pikielny, Barry Dickson, Marc Dionne, James Jepson, Scott Waddell, Stephen Goodwin, Marc Wu, and the Bloomington Stock Centre for fly stocks. Special thanks to all the members of the Gilestro, Dionne and Southall laboratories for constant and invaluable input and support.

## Additional information

### Funding

| Funder | Grant reference number | Author |
|---|---|---|
| Biotechnology and Biological Sciences Research Council | BB/M003930/1 | Alice S French<br>Giorgio F Gilestro |
| European Molecular Biology Organization | ALTF 57-2014 | Esteban J Beckwith |
| H2020 European Research Council | 705930 | Esteban J Beckwith |
| Biotechnology and Biological Sciences Research Council | BB/J014575/1 | Quentin Geissmann<br>Giorgio F Gilestro |

The funders had no role in study design, data collection and interpretation, or the decision to submit the work for publication.

### Author contributions

Esteban J Beckwith, Conceptualization, Investigation, Writing—review and editing; Quentin Geissmann, Software, Visualization, Writing—review and editing; Alice S French, Investigation, Writing—review and editing; Giorgio F Gilestro, Conceptualization, Data curation, Software, Supervision, Funding acquisition, Visualization, Methodology, Writing—original draft, Project administration, Writing—review and editing

### Author ORCIDs

Esteban J Beckwith ⓘ http://orcid.org/0000-0002-3373-1833
Quentin Geissmann ⓘ http://orcid.org/0000-0001-6546-4306
Alice S French ⓘ http://orcid.org/0000-0003-2498-8490
Giorgio F Gilestro ⓘ http://orcid.org/0000-0001-7512-8541

### Decision letter and Author response

Decision letter https://doi.org/10.7554/eLife.27445.023
Author response https://doi.org/10.7554/eLife.27445.024

## Additional files

### Supplementary files

• Supplementary file 1. Details of all statistical comparisons. A text file containing the statistical details of all statistical comparisons.
DOI: https://doi.org/10.7554/eLife.27445.017

• Transparent reporting form
DOI: https://doi.org/10.7554/eLife.27445.018

### Major datasets

The following dataset was generated:

| Author(s) | Year | Dataset title | Dataset URL | Database, license, and accessibility information |
|---|---|---|---|---|
| Giorgio F Gilestro, Esteban J Beckwith, Quentin Geissmann | 2016 | Sample video of Drosophila interaction. Linked to "Beckwith, Geissmann and Gilestro" | http://dx.doi.org/10.5281/zenodo.167551 | Publicly available at Zenodo (https://zenodo.org/). |

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
