## [Decision Letter]

Thank you for submitting your article "Regulation of sleep homeostasis by sexual arousal. Implications for a three-process model of sleep regulation" for consideration by *eLife*. Your article has been favorably evaluated by a Senior Editor and three reviewers, one of whom, Leslie C Griffith (Reviewer #1), is a member of our Board of Reviewing Editors.. The following individual involved in review of your submission has agreed to reveal their identity: Alex C Keene (Reviewer #2).

The reviewers have discussed the reviews with one another and the Reviewing Editor has drafted this decision to help you prepare a revised submission.

Summary:

In this manuscript Beckwith et al. provide evidence that sleep loss can be counteracted by enhancing sexual arousal in male flies, providing a link between sleep homeostasis and sexual arousal. The authors elegantly show that sleep rebound is specifically affected by manipulating pheromonal sensory input, or arousal neurons themselves. The work is innovative because it moves beyond examining sleep in individual flies to address how sleep is modulated by more complex environments. There is growing appreciation throughout the sleep community for the diverse environmental and social factors that impact sleep and this manuscript provides the most in-depth investigation of how social interactions affect sleep to date.

Essential revisions:

1) The major thing that all reviewers agreed needed revision is the focus on "process E". The discussion of “process E” may be better addressed in a review article, or through a research article with greater emphasis on interactions between processes. For example, how does process E affected by process C and S to modulate timing of sleep? Findings here provide the basis for examining these interactions but are insufficient in their current state to propose a novel three process model. The attempts to broadly place the findings into the Borbely model (including the subtitle, which should be changed) distract from the significant scientific findings presented here.

2) Language. Several of the reviewers thought that it was a bit distracting in places.

In the first paragraph of the subsection “Rebound sleep is regulated by species-specific pheromones” it is probably best to remove "conscious" it is never really clearly appropriate. The point can be made just as well without the use of loaded language.

Minimize colloquialisms. Sentences like “At this point of our investigation, we had enough evidence to be convinced…” or “we live in an age of.….” distract from the scientific narrative. In addition, anthropomorphisms like “having experienced the bitterness of rejection.”

Subsection, “Manipulating the extent and nature of sexual arousal”, last paragraph: gender is an anthropomorphism.

[Editors’ note: a previous version of this study was rejected after peer review, but the authors submitted for reconsideration. The first decision letter after peer review is shown below.]

Thank you for submitting your work entitled "Regulation of sleep homeostasis by sex pheromones" for consideration by *eLife*. Your article has been favorably evaluated by a Senior Editor and three reviewers, one of whom, Leslie C Griffith (Reviewer #1), is a member of our Board of Reviewing Editors.

Our decision has been reached after consultation between the reviewers. Based on these discussions and the individual reviews below, we regret to inform you that your work will not be considered further for publication in *eLife*.

This paper is on an incredibly important subject, the regulation of sleep homeostasis. The authors show that there is suppression of rebound sleep after courtship-based sleep deprivation. The initial data seem strong, but the follow up is less convincing on several fronts detailed in the reviews. At the end of the day there is not a convincing mechanistic story here, just a very interesting observation.

*Reviewer #1:*

This paper shows very nicely that sleep deprivation in males induced by courtship of females does not generate significant rebound sleep. The idea that sleep deprivation and homeostatic processes are inviolable has a lot of traction. Reports of the homeostat apparently being "turned off" (migrating birds, postpartum whales etc.) have been dismissed as idiosyncratic to certain species. This report (and to an extent the accompanying paper) finally give us a tractable model system for investigating how the homeostat can be modulated. I think this is an important paper for that reason alone.

That said, there are a few problems with the more mechanistic aspects of the paper. In particular:

1) While the data in Figure 2—figure supplement 1 A, B, C are very compelling, i.e. the differences in the ZT0-3 window are fairly obvious and the onset of sleep is similar for each condition, the data in Figure 2—figure supplement 1 are not. The time course of sleep is altered, not just the amount in that ZT0-3 window. The effect is also very small. If the authors think that it is residual pheromone that is driving the suppression of rebound, a simple manipulation to try is to move the male to a CLEAN tube after the interaction instead of just removing the female.

2) There is also a problem with the data in Figure 2—figure supplement 2. The authors want to make the point that pheromone does not suppress rebound in ppk, but they clearly *do* if you look at a longer time window. By picking ZT0-3 they cut out the main effects which are seen later. I don't think these data are very strong at all.

3) The ppk-GAL4 experiments in Figure 3 are also quite problematic. There are several very important controls missing here: namely the GAL4 and UAS lines alone. Temperature has huge effects on the amount and distribution of sleep. Typically daytime sleep is increased and shifted later in the day and nighttime sleep is suppressed. This is basically what is shown in Figure 3: the total amount of sleep in the 29 °C after sleep deprivation is probably the same (data are not shown) but the peak is shifted. The loss of nighttime sleep is quite profound (though more than one would expect with just temperature in WT) but one needs to see the GAL4 and UAS controls and how they behave at 29 °C to interpret any of these effects. As it stands, I don't think the implication of ppk in this effect is very well supported especially in light of the reported failure of activation of the cells.

4) The focus on ppk (which is not very convincing) has led the authors to ignore some obvious experiments like looking at fru, *Dsx* etc. mutants. If the suppression is sex specific does it require that circuitry?

5) In the Koh paper they show that sleep deprivation can suppress courtship. Yet PRIOR courtship suppresses rebound. This makes no sense – why would a male stop courting an actual female to sleep if he will suppress sleep just due to pheromone reside? Some resolution should be sought on this issue or it at least should be discussed.

*Reviewer #2:*

In this paper, the authors aim to discover new environmental factors affecting sleep rebound, after sleep deprivation, in *Drosophila melanogaster* flies. The authors have designed an elegant set up to measure the activity (or sleep) in individual flies before, during and after their interaction with another fly (interactor). Their important data concern their measures of the sleep profile of exposed (or mock) flies few hours or a day after this interaction. They have manipulated both the interactor fly (with eventual addition of molecules potentially acting as pheromones) and/or the focal male fly by targeted alteration of several properties of his nervous system (associative memory, olfaction, perception of cuticular hydrocarbons).

Although this paper contains a large series of observations, I was not completely convinced by the main conclusions reached by the authors based on the interpretation of the data shown. While they used many state-of-the art genetical tools, I found that the major question and some crucial experiments were missing. Also, some of the statistics seem to be inappropriate.

I have two major concerns:

1) I do not think that the authors showed that female predominant cuticular hydrocarbons (CHCs; 7, 11-heptacosadiene and nonacosadiene) induced a clear effect on sleep rebound for at least two reasons:

Sleep rebound depends on the activity of the focal male fly engaged with the interactor fly. Therefore, with a *D. melanogaster* male or a *D. simulans* female interactor (both flies produce similar CHCs that inhibit *D. melanogaster* male courtship) the focal male will not show an important activity and will be very quickly inhibited by predominant CHCs produced by these interactor flies. This may explain why he shows higher sleep activity during the beginning (ZT0-3) of the rebound day (Figure 1). Differently, with a *D. melanogaster* female the activity of the focal male will be higher than with the two latter interactor flies making possible that he will continue to be excited few hours after the end of interaction (ZT0-3; but however not during the complete day).

The effect shown on Figure 2 is pivotal relatively to the main conclusions of this paper. However, the statistics do not seem to be valid, at least with the elements provided in the stats supplementary files. In the subsection “Statistical analysis and data reproducibility”, the authors wrote "statistical analysis consisted of pairwise Wilcoxon rank sum test with P value adjustment for multiple comparisons (Benjamini and Hochberg, 1995)". However, I believe that while the Wilcoxon test is appropriate to compare matched data, here the sets of data compared two by two are independent. For such comparison, a Mann-Whitney test seems more suitable. This therefore casts doubt on the significance (or non-significance) of the two left cyan box plot bars shown to compare the effect induced by the vehicle control vs. the two 7,11-dienes in CantonS (Figure 2) and ppk23 mutant males (Figure 2).

A Kruskall-Wallis test followed by a post-comparison between pairs of data set (but only in case of a significant difference for the K-W test) would be a reliable complement to test each data set with all the other data sets. This comment also applies for other Figure items (Figure 1, Figure 2 and Figure 3).

2) A more reliable manner to test the effect of female CHCs would consist to use females genetically depleted for the production of these substances. The authors may find plenty of these genotypes after a careful lecture of the literature. Then, after using CHCs depleted females then could add female CHCs on these females, after a mechanical transfer of CHCs of control females, and decipher whether these compounds really affect sleep rebound. Only this type of experiment would provide a decisive experiment to state whether (or not) female 7,11-dienes are involved in the sleep rebound effect. At the moment, any other effect could cause this difference (if significant?) including the difference of activity (see the first point above).

*Reviewer #3:*

In this paper, the authors study the effect of environmental factors on sleep. Sleep loss is induced by pairing a male fly with either a male or a female intruder. The first pairing results in partial sleep loss for the original inhabitant while the presence of a female intruder keeps him up all night. The authors note that male-male paired flies show increased rebound sleep after partial sleep deprivation while male-female paired flies show no rebound sleep, even though they'd been awake for 24 hours. The authors subsequently show that pheromone sensing circuits underlie this decrease in rebound sleep. While these results are interesting, there remain important experiments and clarifications to substantiate the conclusions.

Comments:

1) Do female pheromones affect sleep architecture in isolated males?

2) Rebound sleep can take many forms: increased total sleep is what the authors look at. However, sleep deprivation also decreases brief awakenings, suggesting deeper sleep (Huber, 2004). Likewise, sleep deprivation increases arousal thresholds, resulting in reduced responsiveness to mechanical stimulation (van Alphen, 2013; Faville 2015). Have the authors looked at changes in sleep architecture (for example, increased bout length and decreased bout number or altered arousal thresholds) after social interactions?

3) There are other examples where sleep deprivation doesn't result in rebound sleep – see Paul Shaw's work on the lack of rebound sleep after starvation (Thimgan, 2010) and William Joiner's work on rebound (Seidner, 2015) which showed that homeostatic recovery time can be uncoupled from prior wake time. Seidner et al. also show that many types of neurons can drive wake, but only a few neurons drive sleep homeostasis and that specifically activating cholinergic neurons creates sleep loss followed by strong rebound sleep, activating dopaminergic circuitry causes strong sleep loss that is not followed by rebound and that activating octopaminergic circuitry decreases sleep but also decreases sleep homeostasis (i.e. negative rebound). It would strengthen the paper if the authors discuss their own work in the light of these established findings.

4) The paper relies heavily on the use of 'ethoscopes' to quantify behavior but this method is not properly validated. Referring to an 'in preparation' manuscript is not sufficient for reviewers to comment on the validity of this method.

5) Please show the TrpA1 activation data of ppk23 neurons rather than omitting it.

6) How are the intruder males or females removed? Is it possible that this manipulation stresses the original inhabitant and overrides its rebound sleep?

7) The R2 CaLexa data suggest that flies are still sleep deprived. Would pheromone exposure alone reduce CaLexA activity signals in these neurons thus reducing homeostatic sleep drive?

8) In Figure 3 one cannot exclude the possibility that observed effects are due to temperature rather than Gal4/Shi. The authors should include Gal4/+ and UASshi/+ controls to exclude this possibility. Also, why is rebound only quantified for the first three hours? It seems that ppk25 silencing delays rebound, as the yellow curve is substantially shifted to the right, suggesting that this manipulation delays rebound sleep onset, rather than cancel it out.

9) Materials and methods section, “bootstrap re-sampling with 5000 replicates, was performed in order to generate 95% confidence interval”. Can the authors comment on this procedure and provide either references or validation of this approach?

10) The claim that 'sleep pressure after sleep deprivation can be counteracted by the mere presence of aphrodisiac pheromones' is not fully supported by the data. A direct way of showing this would be to mechanically sleep deprive flies, then present them with pheromones at the onset of rebound sleep and compare rebound sleep amount to vehicle-only controls.

11) Also, if truly counteracting sleep pressure shouldn't the pheromone suppress CaLexA changes in R2? Are the pheromones essentially acting as a sex-specific stimulant?

12) Does pheromone application cancel out rebound sleep or does it delay it? I.e. what happens when the pheromone-saturated paper is removed? Do flies then show an even stronger rebound sleep?

Figure comments:

Figure 1 – Why is rebound sleep only quantified in the first 3 hours? During that period, there is no difference in total sleep between mock and MF (peach) flies. However, Figure 1 suggests that sleep is reduced in MF flies compared to mock flies in the 1-6 (and possibly the 1-12 hour) range.

Figure 1 – the color coding is unclear, making it hard to distinguish between the different behaviors. Please use a larger range of the color palette.

Figure 1—figure supplement 1 what does sleep data look like for undisturbed flies?

---

## [Author Response]

Essential revisions:1) The major thing that all reviewers agreed needed revision is the focus on "process E". The discussion of “process E” may be better addressed in a review article, or through a research article with greater emphasis on interactions between processes. For example, how does process E affected by process C and S to modulate timing of sleep? Findings here provide the basis for examining these interactions but are insufficient in their current state to propose a novel three process model. The attempts to broadly place the findings into the Borbely model (including the subtitle, which should be changed) distract from the significant scientific findings presented here.

We have now removed any mention of process E from the manuscript. We still discuss our findings within the frame of the two process model because we believe this is the right approach but, on reflection, we agree with the reviewers that a proper formalisation of process E may be too premature and misplaced.

2) Language. Several of the reviewers thought that it was a bit distracting in places.In the first paragraph of the subsection “Rebound sleep is regulated by species-specific pheromones” it is probably best to remove "conscious" it is never really clearly appropriate. The point can be made just as well without the use of loaded language.

This correction has now been made.

Minimize colloquialisms. Sentences like “At this point of our investigation, we had enough evidence to be convinced…” or “we live in an age of.….” distract from the scientific narrative. In addition, anthropomorphisms like “having experienced the bitterness of rejection.”

All those corrections were implemented, together with other changes in the same direction.

Subsection, “Manipulating the extent and nature of sexual arousal”, last paragraph: gender is an anthropomorphism.

The sentence was replaced as follows:

“Finally, to investigate whether mere sexual arousal is responsible for this effect, we used flies mutant in the *TDC2* gene, that possess lower levels of tyramine and octopamine (Crocker and Sehgal, 2008) and were previously shown to court male as well as female flies (Huang et al., 2016).”

[Editors’ note: the author responses to the first round of peer review follow.]

Reviewer #1:This paper shows very nicely that sleep deprivation in males induced by courtship of females does not generate significant rebound sleep. The idea that sleep deprivation and homeostatic processes are inviolable has a lot of traction. Reports of the homeostat apparently being "turned off" (migrating birds, postpartum whales etc.) have been dismissed as idiosyncratic to certain species. This report (and to an extent the accompanying paper) finally give us a tractable model system for investigating how the homeostat can be modulated. I think this is an important paper for that reason alone.That said, there are a few problems with the more mechanistic aspects of the paper. In particular:1) While the data in Figure 2—figure supplement 1 A, B, C are very compelling, i.e. the differences in the ZT0-3 window are fairly obvious and the onset of sleep is similar for each condition, the data in Figure 2—figure supplement 1 are not. The time course of sleep is altered, not just the amount in that ZT0-3 window. The effect is also very small. If the authors think that it is residual pheromone that is driving the suppression of rebound, a simple manipulation to try is to move the male to a CLEAN tube after the interaction instead of just removing the female.

The reviewer is absolutely correct. We performed a whole new line of work to explore whether pheromones were the only cue driving the observed effect and we found that it was not the case. In particular, the very experiment suggested by the reviewer is now shown in Figure 6 and goes along with two new experiments in which we modulate sexual arousal (Figure 7 and Figure 8). At the same time, we reinforced the observation that simple exposure to pheromones is sufficient to suppress sleep rebound, performing a new series of experiments shown in Figure 4 and 5. As consequence of all these new interpretations, the focus – and title – of the manuscript have now changed to reflect the role of sexual arousal, not just pheromones.

2) There is also a problem with the data in Figure 2—figure supplement 2. The authors want to make the point that pheromone does not suppress rebound in ppk, but they clearly do if you look at a longer time window. By picking ZT0-3 they cut out the main effects which are seen later. I don't think these data are very strong at all.

The *ppk23*-/- experiment has now moved to its own figure (Figure 3). We believe this presentation better highlights the extent of the effect itself. In ppk23-/-, the presence of pheromone does not suppress rebound in the ZT0-3 window (compare peach with blue in 3B and light-purple to blue in 3C). However, the effect is complicated by the fact that the vehicle itself appear to slightly increase rebound (3A and 3C). On top of this, the *ppk23* mutant experiment is also complicated by the fact that *ppk23* mutant male flies have a reduce courtship to start with. We agree with the reviewer that this this experiment taken alone may at best be suggestive, but definitely not conclusive. However, we believe we now confirm the role of pheromones and PPK23 signalling with two cleaner experiments, shown in Figure 4 and 5. We show that tube change (Figure 4) or TrpA1 activation of ppk23-GAL4 cells (Figure 5) are sufficient to suppress rebound after mechanical sleep deprivation – thus completely removing social interaction from the equation.

3) The ppk-GAL4 experiments in Figure 3 are also quite problematic. There are several very important controls missing here: namely the GAL4 and UAS lines alone. Temperature has huge effects on the amount and distribution of sleep. Typically daytime sleep is increased and shifted later in the day and nighttime sleep is suppressed. This is basically what is shown in Figure 3: the total amount of sleep in the 29 °C after sleep deprivation is probably the same (data are not shown) but the peak is shifted. The loss of nighttime sleep is quite profound (though more than one would expect with just temperature in WT) but one needs to see the GAL4 and UAS controls and how they behave at 29 °C to interpret any of these effects. As it stands, I don't think the implication of ppk in this effect is very well supported especially in light of the reported failure of activation of the cells.

The reviewer was absolutely right. We repeated all those experiments using different thermogenetic lines, after proper clean up by backcrossing several generations and found that all of the shiTS results were indeed artefacts. The effect of temperature was not present on the GAL4 parental line, but it was very strong in the uas-shiTS control line.

We now use uas-TrpA1 in combination with the ppk23 GAL4 driver (Figure 5) and in combination with the split P1-GAL4 driver (Figure 7). In both cases, we include all parental controls and all temperature controls and all lines were backcrossed against *w1118*for five generations.

4) The focus on ppk (which is not very convincing) has led the authors to ignore some obvious experiments like looking at fru, Dsx etc. mutants. If the suppression is sex specific does it require that circuitry?

We now explore the *fru* pathway by using the P1-GAL4 line as a way to induce sexual arousal (Figure 7). We also use two previously described mutants in the tyrosine-tyramine-octopamine pathway to manipulate sexual arousal and place our results in the bigger picture of the sex literature (Figure 8).

5) In the Koh paper they show that sleep deprivation can suppress courtship. Yet PRIOR courtship suppresses rebound. This makes no sense – why would a male stop courting an actual female to sleep if he will suppress sleep just due to pheromone reside? Some resolution should be sought on this issue or it at least should be discussed.

It is difficult for me to comment on this issue at this stage, not having read the Koh paper. However, we manually scored and annotated behaviour of all our paradigmatic interactions with a resolution of 60 seconds, and we clearly see an extremely resilient courtship activity in wild type males, with several instances of repeated copulation (in one case, one male copulates three times in 24 hours with the same female partner). We believe this unprecedented level of sexual activity may partly be due to the fact that flies are housed in small glass tubes, which probably increase their sensory stimulation.

Reviewer #2:[…] Although this paper contains a large series of observations, I was not completely convinced by the main conclusions reached by the authors based on the interpretation of the data shown. While they used many state-of-the art genetical tools, I found that the major question and some crucial experiments were missing. Also, some of the statistics seem to be inappropriate.I have two major concerns:1) I do not think that the authors showed that female predominant cuticular hydrocarbons (CHCs; 7, 11-heptacosadiene and nonacosadiene) induced a clear effect on sleep rebound for at least two reasons:Sleep rebound depends on the activity of the focal male fly engaged with the interactor fly. Therefore, with a D. melanogaster male or a D. simulans female interactor (both flies produce similar CHCs that inhibit D. melanogaster male courtship) the focal male will not show an important activity and will be very quickly inhibited by predominant CHCs produced by these interactor flies. This may explain why he shows higher sleep activity during the beginning (ZT0-3) of the rebound day (Figure 1). Differently, with a D. melanogaster female the activity of the focal male will be higher than with the two latter interactor flies making possible that he will continue to be excited few hours after the end of interaction (ZT0-3; but however not during the complete day).

We agree with the reviewer that the previous conclusions were drawn on an insufficient amount of findings. We now reinforce the role of pheromones through a novel series of experiments (Figure 4 and Figure 5) and expand it through a novel line of hypothesis and experiments (Figure 6, Figure 7, Figure 8). The manuscript has changed considerably thanks to suggestions from the reviewers. Regarding the *D. Simulans* experiment: although we absolutely agree that it is not a conclusive experiment on its own, we still believe it can be used as piece of the puzzle. We now show the behavioural annotation of the *D. melanogaster + D. simulans* MF interaction (Figure 2—figure supplement 2). We also show the annotated high-definition video (Supplementary Video S4). From the behavioural profile it is clear that *D. melanogaster* does show some courtship activity, mainly in the day, but in general the *D. simulans* interaction is more similar to MM interaction that MF. We discuss this in the manuscript too.

The effect shown on Figure 2 is pivotal relatively to the main conclusions of this paper. However, the statistics do not seem to be valid, at least with the elements provided in the stats supplementary files. In the subsection “Statistical analysis and data reproducibility”, the authors wrote "statistical analysis consisted of pairwise Wilcoxon rank sum test with P value adjustment for multiple comparisons (Benjamini and Hochberg, 1995)". However, I believe that while the Wilcoxon test is appropriate to compare matched data, here the sets of data compared two by two are independent. For such comparison, a Mann-Whitney test seems more suitable. This therefore casts doubt on the significance (or non-significance) of the two left cyan box plot bars shown to compare the effect induced by the vehicle control vs. the two 7,11-dienes in CantonS (Figure 2) and ppk23 mutant males (Figure 2).A Kruskall-Wallis test followed by a post-comparison between pairs of data set (but only in case of a significant difference for the K-W test) would be a reliable complement to test each data set with all the other data sets. This comment also applies for other Figure items (Figure 1, Figure 2 and Figure 3).

This comment is conceptually flawless but it highlights a clear misunderstanding: we actually already do use the statistical test suggested by reviewer #2! I believe the confusion springs from the fact that the reviewer believes we used the "Wilcoxon test" (more accurately known as "Wilcoxon signed-rank test"), which is for paired data, while instead we used the "Wilcoxon rank sum test" (aka "Mann-Whitney U test"), which is for two-sample unpaired data. Therefore, we are in fact in full agreement with reviewer #2. Unfortunately, there are many aliases for the same test in the literature (Wilcoxon rank-sum test, Wilcoxon–Mann–Whitney test, Mann–Whitney U test or Mann– Whitney–Wilcoxon test) and we had adopted to use the default name in the statistical package “R”, which is what we used for all data analysis. We now unambiguously clarified our description by stating "statistical analysis consisted of pairwise Wilcoxon rank sum test (i.e. Mann–Whitney U test) with P value adjustment for multiple comparisons (Benjamini and Hochberg, 1995)"

2) A more reliable manner to test the effect of female CHCs would consist to use females genetically depleted for the production of these substances. The authors may find plenty of these genotypes after a careful lecture of the literature. Then, after using CHCs depleted females then could add female CHCs on these females, after a mechanical transfer of CHCs of control females, and decipher whether these compounds really affect sleep rebound. Only this type of experiment would provide a decisive experiment to state whether (or not) female 7,11-dienes are involved in the sleep rebound effect. At the moment, any other effect could cause this difference (if significant?) including the difference of activity (see the first point above).

We agree with the reviewer that this experiment would work too. However, having now moved the focus of the manuscript away from pheromones, we were satisfied by the results shown in Figure 4 and 5.

*Reviewer #3:*

In this paper, the authors study the effect of environmental factors on sleep. Sleep loss is induced by pairing a male fly with either a male or a female intruder. The first pairing results in partial sleep loss for the original inhabitant while the presence of a female intruder keeps him up all night. The authors note that male-male paired flies show increased rebound sleep after partial sleep deprivation while male-female paired flies show no rebound sleep, even though they'd been awake for 24 hours. The authors subsequently show that pheromone sensing circuits underlie this decrease in rebound sleep. While these results are interesting, there remain important experiments and clarifications to substantiate the conclusions.

*Comments:*

1) Do female pheromones affect sleep architecture in isolated males?

We now show the result of this experiment in Figure 4 and the answer is that no, it does not seem to have an effect (gray line to be compared to gray line in C, E: the difference between gray line in G and C is not significant (P=0.2) and neither is the one between 4G and 4E (P=0.09)).

2) Rebound sleep can take many forms: increased total sleep is what the authors look at. However, sleep deprivation also decreases brief awakenings, suggesting deeper sleep (Huber, 2004). Likewise, sleep deprivation increases arousal thresholds, resulting in reduced responsiveness to mechanical stimulation (van Alphen, 2013; Faville 2015). Have the authors looked at changes in sleep architecture (for example, increased bout length and decreased bout number or altered arousal thresholds) after social interactions?

We did look at number of bouts, length of bouts, and latency to first sleep episode in the ZT0-3 interval and in the ZT0-6 interval. We have not seen anything relevant. In particular, number and duration of bouts follow closely the measurement of sleep and it is different between MM and Mock but not different between MF and Mock. We now mention this in the Discussion.

3) There are other examples where sleep deprivation doesn't result in rebound sleep – see Paul Shaw's work on the lack of rebound sleep after starvation (Thimgan, 2010) and William Joiner's work on rebound (Seidner, 2015) which showed that homeostatic recovery time can be uncoupled from prior wake time. Seidner et al. also show that many types of neurons can drive wake, but only a few neurons drive sleep homeostasis and that specifically activating cholinergic neurons creates sleep loss followed by strong rebound sleep, activating dopaminergic circuitry causes strong sleep loss that is not followed by rebound and that activating octopaminergic circuitry decreases sleep but also decreases sleep homeostasis (i.e. negative rebound). It would strengthen the paper if the authors discuss their own work in the light of these established findings.

This a good suggestion. We have added a section in the Discussion.

4) The paper relies heavily on the use of 'ethoscopes' to quantify behavior but this method is not properly validated. Referring to an 'in preparation' manuscript is not sufficient for reviewers to comment on the validity of this method.

The reviewer has another valid point. We now uploaded the ethoscope manuscript on biorxiv http://biorxiv.org/content/early/2017/04/02/113647 and we reference that version throughout the manuscript.

5) Please show the TrpA1 activation data of ppk23 neurons rather than omitting it.

TrpA1 activation of *ppk23* neurons is now the bulk of Figure 5 and Figure 5—figure supplement 1. All thermogenetic experiments were repeated using lines backcrossed 5 times to *w1118*. For good measure, we used lines from our laboratory and lines already backcrossed from James Jepson laboratory (UCL, London). The shi^TS^ result was an artefact of non-backcrossed lines and we have now omitted that part completely.

6) How are the intruder males or females removed? Is it possible that this manipulation stresses the original inhabitant and overrides its rebound sleep?

Animals are removed by tipping the intruder into another tube. It is possible that the procedure is somehow stressful but the MM combination really controls for that, because it is done in the very same way.

7) The R2 CaLexa data suggest that flies are still sleep deprived. Would pheromone exposure alone reduce CaLexA activity signals in these neurons thus reducing homeostatic sleep drive?

CaLexa experiment suggests that MM and MF and MechanicalSD all show the same amount of CaLexA signal. We do not know what this truly means because we do not know enough about the molecular underpinning of this experiment. I think the reviewer is asking what happens at ZT3 in all conditions, which is a very good question. The answer is: we do not know and we cannot really address this issue using CaLexA because of the nature of the GFP based assay: CaLexA can be used to measure build up but not fading out. How sleep pressure dissipates in MF flies is a total new adventure.

8) In Figure 3 one cannot exclude the possibility that observed effects are due to temperature rather than Gal4/Shi. The authors should include Gal4/+ and UASshi/+ controls to exclude this possibility. Also, why is rebound only quantified for the first three hours? It seems that ppk25 silencing delays rebound, as the yellow curve is substantially shifted to the right, suggesting that this manipulation delays rebound sleep onset, rather than cancel it out.

That Figure 3 does not exist anymore, but all thermogenetics experiments now shown in the paper have all the parental and temperature control conditions.

9) Materials and methods section, “bootstrap re-sampling with 5000 replicates, was performed in order to generate 95% confidence interval”. Can the authors comment on this procedure and provide either references or validation of this approach?

All the statistics are done in R using our rethomics software (freely available at https://github.com/gilestrolab/rethomics ). We chose bootstrap c.i. to handle errors in the sleep plots because they provide a very convenient way to visualise linear data.

For a technical explanation of the bootstrap re-sampling c.i. see section 3 (3.3) in [J. Carpenter and J. Bithell, “Bootstrap confidence intervals: when, which, what? A practical guide for medical statisticians,” Statist. Med., vol. 19, no. 9, pp. 1141–1164, May 2000]. We now added this reference to the manuscript.

10) The claim that 'sleep pressure after sleep deprivation can be counteracted by the mere presence of aphrodisiac pheromones' is not fully supported by the data. A direct way of showing this would be to mechanically sleep deprive flies, then present them with pheromones at the onset of rebound sleep and compare rebound sleep amount to vehicle-only controls.

Absolutely right. We now did this and it is shown in the new Figure 4 and 5.

11) Also, if truly counteracting sleep pressure shouldn't the pheromone suppress CaLexA changes in R2? Are the pheromones essentially acting as a sex-specific stimulant?

The reviewer raises a very interesting point about the nature of the molecular correlates of sleep pressure. The honest answer is that we do not know what to hypothesise. A correct answer to this question would require some knowledge of what the CaLexA changes in R2 ultimately mean. As I mentioned before, understanding how sleep pressure dissipates without rebound is a fantastic question that goes well beyond the scope of this work.

12) Does pheromone application cancel out rebound sleep or does it delay it? I.e. what happens when the pheromone-saturated paper is removed? Do flies then show an even stronger rebound sleep?

We looked at sleep for 48 hours after interaction and we never found any sign of rebound. It looks like pheromone application does cancel rebound indeed. We briefly discuss rebound-less sleep in the manuscript.

Figure comments:Figure 1 – Why is rebound sleep only quantified in the first 3 hours? During that period, there is no difference in total sleep between mock and MF (peach) flies.

We focused the quantification on ZT0-3 because this is where we consistently see the biggest difference, in all conditions. It also is the window where the greatest effects of rebound happen after mechanical sleep deprivation, for instance.

However, Figure 1 suggests that sleep is reduced in MF flies compared to mock flies in the 1-6 (and possibly the 1-12 hour) range.

Yes, one may be misled to think so by the graph in Figure 1, but actually analysis of ZT0-6 data show that they would look very similar to ZT0-3. See Author response image 1.

The effect at ZT0-6 is very similar to ZT0-3 for the prototypical experiment in Figure 1 but not for the other conditions, possibly due to the synergy between sexual arousal from past experience and presence of pheromones. We now briefly discuss this synergy in the manuscript. We prefer not to overload the paper with stats for the ZT0-6 window – it would then prompt a whole series of other measurements (e.g. 0-12, 12-24, 0-24, etc.) which are as arbitrary but also, in our opinion, less biologically relevant given the rebound is always best expected in the ZT0-3 window. One aspect to keep in mind is that the use of bootstrap c.i. in ethograms implies that when two lines do not overlap, they may be considered statistically significant in their difference. The readers may be able to better extract information in that way.

Figure 1 – the color coding is unclear, making it hard to distinguish between the different behaviors. Please use a larger range of the color palette.

We now redesigned all figures to use a consistent and enlarged set of colours.

Figure 1—figure supplement 1 what does sleep data look like for undisturbed flies?

This is now shown in Figure 1—figure supplement 1.